# SimPO: Simple Preference Optimization with a Reference-Free Reward

**Yu Meng**[1]*    **Mengzhou Xia**[2]*    **Danqi Chen**[2]

[1]Computer Science Department, University of Virginia
[2]Princeton Language and Intelligence (PLI), Princeton University
`yumeng5@virginia.edu`
`{mengzhou,danqic}@cs.princeton.edu`

## Abstract

Direct Preference Optimization (DPO) is a widely used offline preference optimization algorithm that reparameterizes reward functions in reinforcement learning from human feedback (RLHF) to enhance simplicity and training stability. In this work, we propose SimPO, a simpler yet more effective approach. The effectiveness of SimPO is attributed to a key design: using the *average* log probability of a sequence as the implicit reward. This reward formulation better aligns with model generation and eliminates the need for a reference model, making it more compute and memory efficient. Additionally, we introduce a target reward margin to the Bradley-Terry objective to encourage a larger margin between the winning and losing responses, further improving the algorithm's performance. We compare SimPO to DPO and its recent variants across various state-of-the-art training setups, including both base and instruction-tuned models such as Mistral, Llama 3, and Gemma 2. We evaluate on extensive chat-based evaluation benchmarks, including AlpacaEval 2, MT-Bench, and Arena-Hard. Our results demonstrate that SimPO consistently and significantly outperforms existing approaches without substantially increasing response length. Specifically, SimPO outperforms DPO by up to 6.4 points on AlpacaEval 2 and by up to 7.5 points on Arena-Hard. Our top-performing model, built on Gemma-2-9B-it, achieves a 72.4% length-controlled win rate on AlpacaEval 2, a 59.1% win rate on Arena-Hard, and ranks 1st on Chatbot Arena among <10B models with real user votes.[1]

## 1 Introduction

Learning from human feedback is crucial in aligning large language models (LLMs) with human values and intentions [47], ensuring they are helpful, honest, and harmless [5]. Reinforcement learning from human feedback (RLHF) [18, 58, 68] is a popular method for fine-tuning language models to achieve effective alignment. While the classical RLHF approach [58, 65] has shown impressive results, it presents optimization challenges due to its multi-stage procedure, which involves training a reward model and then optimizing a policy model to maximize that reward [13].

Recently, researchers have been exploring simpler offline algorithms. Direct Preference Optimization (DPO) [61] is one such approach. DPO reparameterizes the reward function in RLHF to directly learn a policy model from preference data, eliminating the need for an explicit reward model. It has gained widespread practical adoption due to its simplicity and stability. In DPO, the implicit reward is formulated using the log ratio of the likelihood of a response between the current policy model and the supervised fine-tuned (SFT) model. However, this reward formulation is not directly aligned with

---

*Equal Contribution.
[1]Code and models can be found at `https://github.com/princeton-nlp/SimPO`.

38th Conference on Neural Information Processing Systems (NeurIPS 2024).

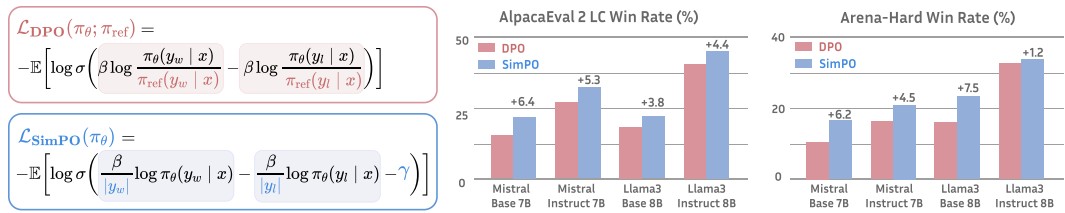

Figure 1: SimPO and DPO mainly differ in their reward formulation, as indicated in the shaded box. SimPO outperforms DPO significantly across a range of settings on AlpacaEval 2 and Arena-Hard.

the metric used to guide generation, which is approximately the average log likelihood of a response generated by the policy model. We hypothesize that this discrepancy between training and inference may lead to suboptimal performance.

In this work, we propose SimPO, a simple yet effective offline preference optimization algorithm (Figure 1). The core of our algorithm aligns the reward function in the preference optimization objective with the generation metric. SimPO consists of two major components: (1) a length-normalized reward, calculated as the *average* log probability of all tokens in a response using the policy model, and (2) a target reward margin to ensure the reward difference between winning and losing responses exceeds this margin. In summary, SimPO has the following properties:

Table 1: Length-controlled (LC) and raw win rate (WR), and generation lengths of top models on the AlpacaEval 2 Leaderboard. **Bold** are the models we trained.

| Model | LC (%) | WR (%) | Len. |
|---|---|---|---|
| **Gemma-2-9B-it-SimPO** | 72.4 | 65.9 | 1833 |
| GPT-4 Turbo (04/09) | 55.0 | 46.1 | 1802 |
| Gemma-2-9B-it | 51.1 | 38.1 | 1571 |
| **Llama-3-8B-Instruct-SimPO** | 44.7 | 40.5 | 1825 |
| Claude 3 Opus | 40.5 | 29.1 | 1388 |
| **Llama-3-8B-Instruct-DPO** | 40.3 | 37.9 | 1837 |
| Llama-3-70B-Instruct | 34.4 | 33.2 | 1919 |
| Llama-3-8B-Instruct | 26.0 | 25.3 | 1899 |

- **Simplicity**: SimPO does not require a reference model, making it more lightweight and easier to implement compared to DPO and other reference-based methods.

- **Significant performance advantage**: Despite its simplicity, SimPO significantly outperforms DPO and its latest variants (*e.g.*, a recent reference-free objective ORPO [38]). The performance advantage is consistent across various training setups and extensive chat-based evaluations, including AlpacaEval 2 [51, 28] and the challenging Arena-Hard [50] benchmark. It achieves up to a 6.4 point improvement on AlpacaEval 2 and a 7.5 point improvement on Arena-Hard compared to DPO (Figure 1).

- **Minimal length exploitation**: SimPO does not significantly increase response length compared to the SFT or DPO models (Table 1), indicating minimal length exploitation [28, 66, 80].

Extensive analysis shows that SimPO utilizes preference data more effectively, leading to a more accurate likelihood ranking of winning and losing responses on a held-out validation set, which in turn translates to a better policy model. As shown in Table 1, our Gemma-2-9B-it-SimPO model achieves state-of-the-art performance, with a 72.4% length-controlled win rate on AlpacaEval 2 and a 59.1% win rate on Arena-Hard, establishing it as the strongest open-source model under 10B parameters. Most notably, when evaluated on Chatbot Arena [17] with real user votes, our model significantly improved upon the initial Gemma-2-9B-it model, advancing from 36th to 25th place and ranking first among all <10B models on the leaderboard.[2]

## 2 SimPO: Simple Preference Optimization

In this section, we first introduce the background of DPO (§2.1). Then we identify the discrepancy between DPO's reward and the likelihood metric used for generation, and propose an alternative reference-free reward formulation that mitigates this issue (§2.2). Finally, we derive the SimPO objective by incorporating a target reward margin term into the Bradley-Terry model (§2.3).

---

[2]As of September 16th, 2024.

## 2.1 Background: Direct Preference Optimization (DPO)

DPO [61] is one of the most popular preference optimization methods. Instead of learning an explicit reward model [58], DPO reparameterizes the reward function $r$ using a closed-form expression with the optimal policy:

$$r(x, y) = \beta \log \frac{\pi_\theta(y \mid x)}{\pi_{\text{ref}}(y \mid x)} + \beta \log Z(x), \tag{1}$$

where $\pi_\theta$ is the policy model, $\pi_{\text{ref}}$ is the reference policy, typically the supervised fine-tuned (SFT) model, and $Z(x)$ is the partition function. By incorporating this reward formulation into the Bradley-Terry (BT) ranking objective [11], $p(y_w \succ y_l \mid x) = \sigma\left(r(x, y_w) - r(x, y_l)\right)$, DPO expresses the probability of preference data with the policy model rather than the reward model, yielding the following objective:

$$\mathcal{L}_{\text{DPO}}(\pi_\theta; \pi_{\text{ref}}) = -\mathbb{E}_{(x, y_w, y_l) \sim \mathcal{D}} \left[ \log \sigma \left( \beta \log \frac{\pi_\theta(y_w \mid x)}{\pi_{\text{ref}}(y_w \mid x)} - \beta \log \frac{\pi_\theta(y_l \mid x)}{\pi_{\text{ref}}(y_l \mid x)} \right) \right], \tag{2}$$

where $(x, y_w, y_l)$ are preference pairs consisting of the prompt, the winning response, and the losing response from the preference dataset $\mathcal{D}$.

## 2.2 A Simple Reference-Free Reward Aligned with Generation

**Discrepancy between reward and generation for DPO.** Using Eq. (1) as the implicit reward has the following drawbacks: (1) it requires a reference model $\pi_{\text{ref}}$ during training, which incurs additional memory and computational costs; and (2) it creates a mismatch between the reward optimized in training and the log-likelihood optimized during inference, where no reference model is involved. This means that in DPO, for any triple $(x, y_w, y_l)$, satisfying the reward ranking $r(x, y_w) > r(x, y_l)$ does not necessarily mean that the likelihood ranking $p_\theta(y_w \mid x) > p_\theta(y_l \mid x)$ is met (here $p_\theta$ is the average log-likelihood in Eq. (3)). In our experiments, we observed that only $\sim 50\%$ of the triples from the training set satisfy this condition when trained with DPO (Figure 4b). This observation aligns with a concurrent work [14], which finds that existing models trained with DPO exhibit random ranking accuracy in terms of average log-likelihood, even after extensive preference optimization.

**Length-normalized reward formulation.** One solution is to use the *summed* token log probability as the reward, but this suffers from *length bias*–longer sequences tend to have lower log probabilities. Consequently, when $y_w$ is longer than $y_l$, optimizing the summed log probability as a reward forces the model to artificially inflate probabilities for longer sequences to ensure $y_w$ receives a higher reward than $y_l$. This overcompensation increases the risk of degeneration. To address this issue, we consider using the *average* log-likelihood as the implicit reward:

$$p_\theta(y \mid x) = \frac{1}{|y|} \log \pi_\theta(y \mid x) = \frac{1}{|y|} \sum_{i=1}^{|y|} \log \pi_\theta(y_i \mid x, y_{<i}). \tag{3}$$

This metric is commonly used for ranking options in beam search [33, 49] and multiple-choice tasks within language models [12, 37, 58]. Naturally, we consider replacing the reward formulation in DPO with $p_\theta$ in Eq. (3), so that it aligns with the likelihood metric that guides generation. This results in a length-normalized reward:

$$r_{\text{SimPO}}(x, y) = \frac{\beta}{|y|} \log \pi_\theta(y \mid x) = \frac{\beta}{|y|} \sum_{i=1}^{|y|} \log \pi_\theta(y_i \mid x, y_{<i}), \tag{4}$$

where $\beta$ is a constant that controls the scaling of the reward difference. We find that normalizing the reward with response lengths is crucial; removing the length normalization term from the reward formulation results in a bias toward generating longer but lower-quality sequences (see Section 4.4 for more details). Consequently, this reward formulation eliminates the need for a reference model, enhancing memory and computational efficiency compared to reference-dependent algorithms.

## 2.3 The SimPO Objective

**Target reward margin.** Additionally, we introduce a target reward margin term, $\gamma > 0$, to the Bradley-Terry objective to ensure that the reward for the winning response, $r(x, y_w)$, exceeds the

reward for the losing response, $r(x, y_l)$, by at least $\gamma$:

$$p(y_w \succ y_l \mid x) = \sigma\left(r(x, y_w) - r(x, y_l) - \gamma\right). \tag{5}$$

The margin between two classes is known to influence the generalization capabilities of classifiers [1, 10, 22, 30].[3] In standard training settings with random model initialization, increasing the target margin typically improves generalization. In preference optimization, the two classes are the winning and losing responses for a single input. In practice, we observe that generation quality initially improves with an increasing target margin but degrades when the margin becomes too large (§4.3). One of DPO's variants, IPO [6], also formulates a target reward margin similar to SimPO. However, its full objective is not as effective as SimPO (§4.1).

**Objective.**    Finally, we obtain the SimPO objective by plugging Eq. (4) into Eq. (5):

$$\mathcal{L}_{\text{SimPO}}(\pi_\theta) = -\mathbb{E}_{(x,y_w,y_l)\sim\mathcal{D}}\left[\log\sigma\left(\frac{\beta}{|y_w|}\log\pi_\theta(y_w|x) - \frac{\beta}{|y_l|}\log\pi_\theta(y_l|x) - \gamma\right)\right]. \tag{6}$$

In summary, SimPO employs an implicit reward formulation that directly aligns with the generation metric, eliminating the need for a reference model. Additionally, it introduces a target reward margin $\gamma$ to help separating the winning and losing responses. In Appendix F, we provide a gradient analysis of SimPO and DPO to further understand the differences between the two methods.

**Preventing catastrophic forgetting without KL regularization.**    Although SimPO does not impose KL regularization, we find that a combination of practical factors ensures effective learning from preference data while maintaining generalization, leading to an empirically low KL divergence from the reference model. These factors are: (1) a small learning rate, (2) a preference dataset that covers diverse domains and tasks, and (3) the intrinsic robustness of LLMs to learn from new data without forgetting prior knowledge. We present KL divergence experiments in Section 4.4.

# 3    Experimental Setup

**Models and training settings.**    We perform preference optimization with two families of models, Llama-3-8B [2] and Mistral-7B [40], under two setups: Base and Instruct. In this section, our goal is to understand the performance of SimPO vs. other preference optimization methods in different experimental setups. Our strongest model is based on Gemma-2-9B (Instruct setup) with a stronger reward model, RLHFlow/ArmoRM-Llama3-8B-v0.1 [79] (Table 1). We will present and discuss these results in Appendix J.

For the **Base** setup, we follow the training pipeline of Zephyr [75]. First, we train a base model (*i.e.*, mistralai/Mistral-7B-v0.1, or meta-llama/Meta-Llama-3-8B) on the UltraChat-200k dataset [25] to obtain an SFT model. Then, we perform preference optimization on the UltraFeedback dataset [23] using the SFT model as the starting point. This setup provides *a high level of transparency*, as the SFT models are trained on open-source data.

For the **Instruct** setup, we use an off-the-shelf instruction-tuned model (*i.e.*, meta-llama/Meta-Llama-3-8B-Instruct, or mistralai/Mistral-7B-Instruct-v0.2) as the SFT models.[4] These models have undergone extensive instruction-tuning processes, making them more powerful and robust than the SFT models in the Base setup. However, they are also *more opaque* because their RLHF procedure is not publicly disclosed. To mitigate the distribution shift between SFT models and the preference optimization process, we generate the preference dataset using the SFT models following [74]. This makes our Instruct setup closer to an *on-policy* setting. Specifically, we use prompts from the UltraFeedback dataset and regenerate the chosen and rejected response pairs $(y_w, y_l)$ with the SFT models. For each prompt $x$, we generate 5 responses using the SFT model with a sampling temperature of 0.8. We then use llm-blender/PairRM [41] to score the 5 responses, selecting the

---

[3]This margin is termed *home advantage* in Bradley-Terry models [1, 30].

[4]It is unclear whether the released instruct checkpoints have undergone supervised fine-tuning (SFT) or the complete RLHF pipeline. For simplicity, we refer to these checkpoints as SFT models.

Table 2: Evaluation details for AlpacaEval 2 [51], Arena-Hard [50], and MT-Bench [94]. The baseline model refers to the model compared against. GPT-4 Turbo corresponds to GPT-4-Preview-1106.

| | # Exs. | Baseline Model | Judge Model | Scoring Type | Metric |
|---|---|---|---|---|---|
| **AlpacaEval 2** | 805 | GPT-4 Turbo | GPT-4 Turbo | Pairwise comparison | LC & raw win rate |
| **Arena-Hard** | 500 | GPT-4-0314 | GPT-4 Turbo | Pairwise comparison | Win rate |
| **MT-Bench** | 80 | - | GPT-4/GPT-4 Turbo | Single-answer grading | Rating of 1-10 |

highest-scoring one as $y_w$ and the lowest-scoring one as $y_l$. We only generated data in a single pass instead of iteratively as in [74].[5]

In summary, we have four setups: Llama-3-Base, Llama-3-Instruct, Mistral-Base, and Mistral-Instruct. We believe these configurations represent the state-of-the-art, placing our models among the top performers on various leaderboards. We encourage future research to adopt these settings for better and fairer comparisons of different algorithms. Additionally, we find that tuning hyperparameters is crucial for achieving optimal performance with all the offline preference optimization algorithms, including DPO and SimPO. Generally, for SimPO, setting $\beta$ between 2.0 and 2.5 and $\gamma$ between 0.5 and 1.5 leads to good performance across all setups. For more details, please refer to Appendix B.

**Evaluation benchmarks.** We primarily assess our models using three of the most popular open-ended instruction-following benchmarks: MT-Bench [94], AlpacaEval 2 [51], and Arena-Hard v0.1 [50]. These benchmarks evaluate the models' versatile conversational abilities across a diverse set of queries and have been widely adopted by the community (details in Table 2). AlpacaEval 2 consists of 805 questions from 5 datasets, and MT-Bench covers 8 categories with 80 questions. The most recently released Arena-Hard is an enhanced version of an MT-Bench, incorporating 500 well-defined technical problem-solving queries. We report scores following each benchmark's evaluation protocol. For AlpacaEval 2, we report both the raw win rate (WR) and the length-controlled win rate (LC) [28]. The LC metric is specifically designed to be robust against model verbosity. For Arena-Hard, we report the win rate (WR) against the baseline model. For MT-Bench, we report the average MT-Bench score with GPT-4 and GPT-4-Preview-1106 as the judge model.[6] For decoding details, please refer to Appendix B. We also evaluate on downstream tasks from the Huggingface Open Leaderboard benchmarks [9], with additional details in in Appendix C.

**Baselines.** We compare SimPO with other *offline* preference optimization methods listed in Table 3.[7] RRHF [86] and SLiC-HF [91] are ranking losses. RRHF uses length-normalized log-likelihood, similar to SimPO's reward function, while SLiC-HF uses log-likelihood directly and includes an SFT objective. IPO [6] is a theoretically grounded approach method that avoids DPO's assumption that pairwise preferences can be replaced with pointwise rewards. CPO [83] uses sequence likelihood as a reward and trains alongside an SFT objective. KTO [29] learns from non-paired preference data.

Table 3: Various preference optimization objectives given preference data $\mathcal{D} = (x, y_w, y_l)$, where $x$ is an input, and $y_w$ and $y_l$ are the winning and losing responses.

| Method | Objective |
|---|---|
| RRHF [86] | $\max\left(0, -\frac{1}{\lvert y_w\rvert}\log\pi_\theta(y_w\lvert x) + \frac{1}{\lvert y_l\rvert}\log\pi_\theta(y_l\lvert x)\right) - \lambda\log\pi_\theta(y_w\lvert x)$ |
| SLiC-HF [91] | $\max\left(0, \delta - \log\pi_\theta(y_w\lvert x) + \log\pi_\theta(y_l\lvert x)\right) - \lambda\log\pi_\theta(y_w\lvert x)$ |
| DPO [61] | $-\log\sigma\left(\beta\log\frac{\pi_\theta(y_w\lvert x)}{\pi_{\text{ref}}(y_w\lvert x)} - \beta\log\frac{\pi_\theta(y_l\lvert x)}{\pi_{\text{ref}}(y_l\lvert x)}\right)$ |
| IPO [6] | $\left(\log\frac{\pi_\theta(y_w\lvert x)}{\pi_{\text{ref}}(y_w\lvert x)} - \log\frac{\pi_\theta(y_l\lvert x)}{\pi_{\text{ref}}(y_l\lvert x)} - \frac{1}{2\tau}\right)^2$ |
| CPO [83] | $-\log\sigma\left(\beta\log\pi_\theta(y_w\lvert x) - \beta\log\pi_\theta(y_l\lvert x)\right) - \lambda\log\pi_\theta(y_w\lvert x)$ |
| KTO [29] | $-\lambda_w\sigma\left(\beta\log\frac{\pi_\theta(y_w\lvert x)}{\pi_{\text{ref}}(y_w\lvert x)} - z_{\text{ref}}\right) + \lambda_l\sigma\left(z_{\text{ref}} - \beta\log\frac{\pi_\theta(y_l\lvert x)}{\pi_{\text{ref}}(y_l\lvert x)}\right)$, where $z_{\text{ref}} = \mathbb{E}_{(x,y)\sim\mathcal{D}}\left[\beta\text{KL}\left(\pi_\theta(y\lvert x)\|\pi_{\text{ref}}(y\lvert x)\right)\right]$ |
| ORPO [38] | $-\log p_\theta(y_w\lvert x) - \lambda\log\sigma\left(\log\frac{p_\theta(y_w\lvert x)}{1-p_\theta(y_w\lvert x)} - \log\frac{p_\theta(y_l\lvert x)}{1-p_\theta(y_l\lvert x)}\right)$, where $p_\theta(y\lvert x) = \exp\left(\frac{1}{\lvert y\rvert}\log\pi_\theta(y\lvert x)\right)$ |
| R-DPO [60] | $-\log\sigma\left(\beta\log\frac{\pi_\theta(y_w\lvert x)}{\pi_{\text{ref}}(y_w\lvert x)} - \beta\log\frac{\pi_\theta(y_l\lvert x)}{\pi_{\text{ref}}(y_l\lvert x)} + (\alpha\lvert y_w\rvert - \alpha\lvert y_l\rvert)\right)$ |
| **SimPO** | $-\log\sigma\left(\frac{\beta}{\lvert y_w\rvert}\log\pi_\theta(y_w\lvert x) - \frac{\beta}{\lvert y_l\rvert}\log\pi_\theta(y_l\lvert x) - \gamma\right)$ |

---

[5]We also experimented with using a stronger reward model, RLHFlow/ArmoRM-Llama3-8B-v0.1 [79], to rank generated data, which yields significantly improved performance (see Appendix H and Appendix J). This is the reward model we used in our Gemma 2 experiments.

[6]GPT-4-Preview-1106 produces more accurate reference answers and judgments compared to GPT-4.

[7]Many recent studies [85, 69] have extensively compared DPO and PPO [65]. We will leave the comparison of PPO and SimPO to future work.

Table 4: AlpacaEval 2 [51], Arena-Hard [50], and MT-Bench [94] results under the four settings. LC and WR denote length-controlled and raw win rate, respectively. We train SFT models for Base settings on the UltraChat dataset. For Instruct settings, we use off-the-shelf models as the SFT model.

| Method | Mistral-Base (7B) | | | | | Mistral-Instruct (7B) | | | | |
| | AlpacaEval 2 | | Arena-Hard | MT-Bench | | AlpacaEval 2 | | Arena-Hard | MT-Bench | |
| | LC (%) | WR (%) | WR (%) | GPT-4 Turbo | GPT-4 | LC (%) | WR (%) | WR (%) | GPT-4 Turbo | GPT-4 |
|---|---|---|---|---|---|---|---|---|---|---|
| SFT | 8.4 | 6.2 | 1.3 | 4.8 | 6.3 | 17.1 | 14.7 | 12.6 | 6.2 | 7.5 |
| RRHF [86] | 11.6 | 10.2 | 5.8 | 5.4 | 6.7 | 25.3 | 24.8 | 18.1 | 6.5 | 7.6 |
| SLiC-HF [91] | 10.9 | 8.9 | 7.3 | 5.8 | **7.4** | 24.1 | 24.6 | 18.9 | 6.5 | **7.8** |
| DPO [61] | 15.1 | 12.5 | 10.4 | 5.9 | 7.3 | 26.8 | 24.9 | 16.3 | 6.3 | 7.6 |
| IPO [6] | 11.8 | 9.4 | 7.5 | 5.5 | 7.2 | 20.3 | 20.3 | 16.2 | 6.4 | **7.8** |
| CPO [83] | 9.8 | 8.9 | 6.9 | 5.4 | 6.8 | 23.8 | 28.8 | **22.6** | 6.3 | 7.5 |
| KTO [29] | 13.1 | 9.1 | 5.6 | 5.4 | 7.0 | 24.5 | 23.6 | 17.9 | 6.4 | 7.7 |
| ORPO [38] | 14.7 | 12.2 | 7.0 | 5.8 | 7.3 | 24.5 | 24.9 | 20.8 | 6.4 | 7.7 |
| R-DPO [60] | 17.4 | 12.8 | 8.0 | 5.9 | **7.4** | 27.3 | 24.5 | 16.1 | 6.2 | 7.5 |
| SimPO | **21.5** | **20.8** | **16.6** | **6.0** | 7.3 | **32.1** | **34.8** | 21.0 | **6.6** | 7.6 |

| Method | Llama-3-Base (8B) | | | | | Llama-3-Instruct (8B) | | | | |
| | AlpacaEval 2 | | Arena-Hard | MT-Bench | | AlpacaEval 2 | | Arena-Hard | MT-Bench | |
| | LC (%) | WR (%) | WR (%) | GPT-4 Turbo | GPT-4 | LC (%) | WR (%) | WR (%) | GPT-4 Turbo | GPT-4 |
|---|---|---|---|---|---|---|---|---|---|---|
| SFT | 6.2 | 4.6 | 3.3 | 5.2 | 6.6 | 26.0 | 25.3 | 22.3 | 6.9 | 8.1 |
| RRHF [86] | 12.1 | 10.1 | 6.3 | 5.8 | 7.0 | 31.3 | 28.4 | 26.5 | 6.7 | 7.9 |
| SLiC-HF [91] | 12.3 | 13.7 | 6.0 | 6.3 | 7.6 | 26.9 | 27.5 | 26.2 | 6.8 | 8.1 |
| DPO [61] | 18.2 | 15.5 | 15.9 | 6.5 | 7.7 | 40.3 | 37.9 | 32.6 | **7.0** | 8.0 |
| IPO [6] | 14.4 | 14.2 | 17.8 | 6.5 | 7.4 | 35.6 | 35.6 | 30.5 | **7.0** | **8.3** |
| CPO [83] | 10.8 | 8.1 | 5.8 | 6.0 | 7.4 | 28.9 | 32.2 | 28.8 | **7.0** | 8.0 |
| KTO [29] | 14.2 | 12.4 | 12.5 | 6.3 | **7.8** | 33.1 | 31.8 | 26.4 | 6.9 | 8.2 |
| ORPO [38] | 12.2 | 10.6 | 10.8 | 6.1 | 7.6 | 28.5 | 27.4 | 25.8 | 6.8 | 8.0 |
| R-DPO [60] | 17.6 | 14.4 | 17.2 | **6.6** | 7.5 | 41.1 | 37.8 | 33.1 | **7.0** | 8.0 |
| SimPO | **22.0** | **20.3** | **23.4** | **6.6** | 7.7 | **44.7** | **40.5** | **33.8** | **7.0** | 8.0 |

ORPO [38][8] introduces a reference-model-free odd ratio term to directly contrast winning and losing responses with the policy model and jointly trains with the SFT objective. R-DPO [60] is a modified version of DPO that includes an additional regularization term to prevent exploitation of length. We thoroughly tune the hyperparameters for each baseline and report the best performance. We find that *many variants of DPO do not empirically present an advantage over standard DPO*. Further details can be found in Appendix B.

## 4 Experimental Results

In this section, we present main results of our experiments, highlighting the superior performance of SimPO on various benchmarks and ablation studies (§4.1). We provide an in-depth understanding of the following components: (1) length normalization (§4.2), (2) the margin term $\gamma$ (§4.3), and (3) why SimPO outperforms DPO (§4.4). Unless otherwise specified, the ablation studies are conducted using the Mistral-Base setting.

### 4.1 Main Results and Ablations

**SimPO consistently and significantly outperforms existing preference optimization methods.** As shown in Table 4, while all preference optimization algorithms enhance performance over the SFT model, SimPO, despite its simplicity, achieves the best overall performance across all benchmarks and settings. These consistent and significant improvements highlight the robustness and effectiveness of SimPO. Notably, SimPO outperforms the best baseline by 3.6 to 4.8 points on the AlpacaEval 2 LC win rate across various settings. On Arena-Hard, SimPO consistently achieves superior performance,

---

[8]ORPO can directly train on preference data without the SFT stage. For fair comparisons, we start ORPO from the same SFT checkpoints as other baselines, which yields better results than starting from base checkpoints.

Table 5: Ablation studies under Mistral-Base and Mistral-Instruct settings. We ablate each key design of SimPO: (1) removing length normalization in Eq. (4) (*i.e.*, w/o LN); (2) setting target reward margin $\gamma$ to be 0 in Eq. (6) (*i.e.*, $\gamma = 0$).

| Method | Mistral-Base (7B) Setting | | | | | Mistral-Instruct (7B) Setting | | | | |
| | AlpacaEval 2 | | Arena-Hard | MT-Bench | | AlpacaEval 2 | | Arena-Hard | MT-Bench | |
| | LC (%) | WR (%) | WR (%) | GPT-4 Turbo | GPT-4 | LC (%) | WR (%) | WR (%) | GPT-4 Turbo | GPT-4 |
|---|---|---|---|---|---|---|---|---|---|---|
| DPO | 15.1 | 12.5 | 10.4 | 5.9 | 7.3 | 26.8 | 24.9 | 16.3 | 6.3 | 7.6 |
| SimPO | 21.5 | 20.8 | 16.6 | 6.0 | 7.3 | 32.1 | 34.8 | 21.0 | 6.6 | 7.6 |
| w/o LN | 11.9 | 13.2 | 9.4 | 5.5 | 7.3 | 19.1 | 19.7 | 16.3 | 6.4 | 7.6 |
| $\gamma = 0$ | 16.8 | 14.3 | 11.7 | 5.6 | 6.9 | 30.9 | 34.2 | 20.5 | 6.6 | 7.7 |

(a) Reward optimization.   (b) SimPO.   (c) SimPO without LN.

Figure 2: Effect of length normalization (LN). (a) Relationship between reward margin and length difference between winning and losing responses. (b) Spearman correlation between average log probability and response length for SimPO. (c) Spearman correlation for SimPO without LN.

though it is occasionally surpassed by CPO [83]. We find that CPO generates responses that are, on average, 50% longer than those generated by SimPO (See Table 10). Arena-Hard might favor longer generations due to the absence of a length penalty in its evaluation.

**Benchmark quality varies.**   Although all three benchmarks are widely adopted, we find that MT-Bench exhibits poor separability across different methods. Minor differences between methods on MT-Bench may be attributed to randomness, likely due to the limited scale of its evaluation data and its single-instance scoring protocol. This finding aligns with observations reported in [50]. In contrast, AlpacaEval 2 and Arena-Hard provide more meaningful distinctions between different methods. We observe that the win rate on Arena-Hard is significantly lower than on AlpacaEval 2, indicating that Arena-Hard is a more challenging benchmark.[9]

**The *Instruct* setting introduces significant performance gains.**   Across all benchmarks, we observe that the *Instruct* setting consistently outperforms the *Base* setting. This improvement is likely due to the higher quality of SFT models used for initialization and the generation of more high-quality preference data by these models.

**Both key designs in SimPO are crucial.**   In Table 5, we demonstrate results from ablating each key design of SimPO: (1) removing length normalization in Eq. (4) (*i.e.*, w/o LN); (2) setting the target reward margin to be 0 in Eq. (6) (*i.e.*, $\gamma = 0$). Removing the length normalization has the most negative impact on the results. Our examination reveals that this leads to the generation of long and repetitive patterns, substantially degrading the overall quality of the output (See Appendix E). Setting $\gamma$ to 0 yields also leads to a performance degradation compared to SimPO, indicating that it is not the optimal target reward margin. In the following subsections, we conduct in-depth analyses to better understand both design choices.

## 4.2   Length Normalization (LN) Prevents Length Exploitation

**LN leads to an increase in the reward difference for all preference pairs, regardless of their length.**   The Bradley-Terry objective in Eq. (5) essentially aims to optimize the reward difference

---

[9]Although our models excel on benchmarks, these evaluations have limitations, including restricted query space and potential biases from model-based evaluations. Efforts like WildBench [88] aim to expand these spaces, where SimPO models demonstrate competitive performance.

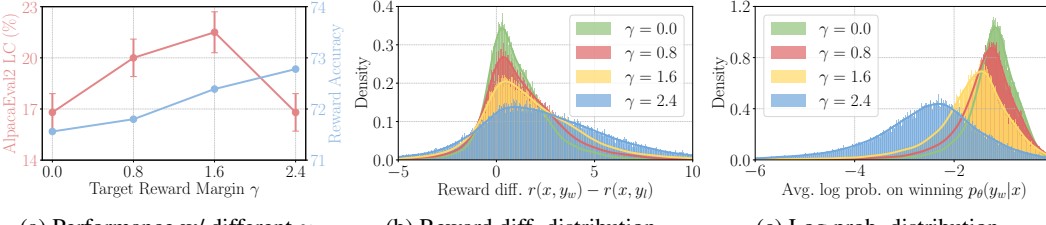

| (a) Performance w/ different $\gamma$. | (b) Reward diff. distribution. | (c) Log prob. distribution. |

Figure 3: Study of the margin $\gamma$. (a) Reward accuracy and AlpacaEval2 LC win rate under different $\gamma$ values. (b) Reward difference distribution under different $\gamma$ values. (c) Log likelihood distribution on chosen responses under different $\gamma$ values.

$\Delta r = r(x, y_w) - r(x, y_l)$ to exceed the target margin $\gamma$. We investigate the relationship between the learned reward differences and the length difference $\Delta l = |y_w| - |y_l|$ between the winning and losing responses from the training set of UltraFeedback. We measure the difference of reward ($r_{\text{SimPO}}$; Eq. (4)) using the SFT model, the SimPO model, and a model trained with SimPO but without length normalization. We present the results in Figure 2a and observe that SimPO with LN consistently achieves a positive reward margin for all response pairs, regardless of their length difference, and consistently improves the margin over the SFT model. In contrast, SimPO without LN results in a negative reward difference for preference pairs when the winning response is shorter than the losing response, indicating that the model learns poorly for these instances.

**Removing LN results in a strong positive correlation between the reward and response length, leading to length exploitation.** Figures 2b and 2c illustrate the average log likelihood ($p_\theta$ in Eq. (3)) versus response length on a held-out set for models trained with SimPO and SimPO without LN. The model trained without LN exhibits a much stronger positive Spearman correlation between likelihood and response length compared to SimPO, indicating a tendency to exploit length bias and generate longer sequences (see Appendix E). In contrast, SimPO results in a Spearman correlation coefficient similar to the SFT model (see Figure 6a).

## 4.3   The Impact of Target Reward Margin in SimPO

**Influence of $\gamma$ on reward accuracy and win rate.** We investigate how the target reward margin $\gamma$ in SimPO affects the reward accuracy on a held-out set and win rate on AlpacaEval 2, presenting the results in Figure 3a. Reward accuracy is measured as the percentage of preference pairs where the winning response ends up having a higher reward for the winning response than the losing response (*i.e.*, $r(x, y_w) > r(x, y_l)$). We observe that reward accuracy increases with $\gamma$ on both benchmarks, indicating that enforcing a larger target reward margin effectively improves reward accuracy. However, the win rate on AlpacaEval 2 first increases and then decreases with $\gamma$, suggesting that generation quality is not solely determined by the reward margin.

**Impact of $\gamma$ on the reward distribution.** We visualize the distribution of the learned reward margin $r(x, y_w) - r(x, y_l)$ and the reward of winning responses $r(x, y_w)$ under varying $\gamma$ values in Figure 2b and Figure 2c. Notably, increasing $\gamma$ tends to flatten both distributions and reduce the average log likelihood of winning sequences. This initially improves performance but can eventually lead to model degeneration. We hypothesize that there is a trade-off between accurately approximating the true reward distribution and maintaining a well-calibrated likelihood when setting the $\gamma$ value. Further exploration of this balance is deferred to future work.

## 4.4   In-Depth Analysis of DPO vs. SimPO

In this section, we compare SimPO to DPO in terms of (1) likelihood-length correlation, (2) reward formulation, (3) reward accuracy, and (4) algorithm efficiency. We demonstrate that SimPO outperforms DPO in terms of reward accuracy and efficiency.

**DPO reward implicitly facilitates length normalization.** Although the DPO reward expression $r(x, y) = \beta \log \frac{\pi_\theta(y|x)}{\pi_{\text{ref}}(y|x)}$ (with the partition function excluded) lacks an explicit term for length normalization, the logarithmic ratio between the policy model and the reference model can serve to

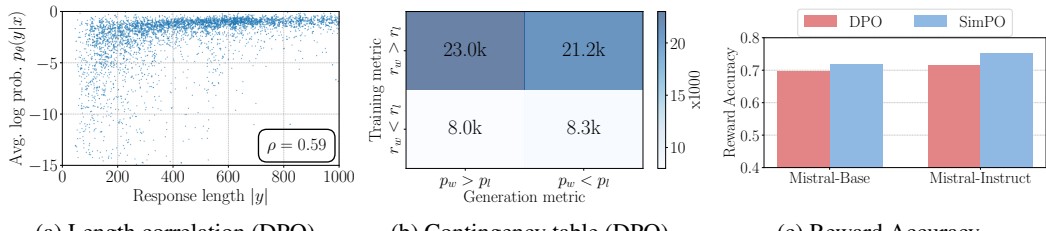

(a) Length correlation (DPO). (b) Contingency table (DPO). (c) Reward Accuracy.

Figure 4: Comparison between SimPO and DPO, measured on UltraFeedback. (a) Spearman correlation between average log probability and response length for DPO. (b) Contingency table of rankings based on DPO rewards and the average log likelihood (measured on the training set). (c) Reward accuracy of DPO and SimPO.

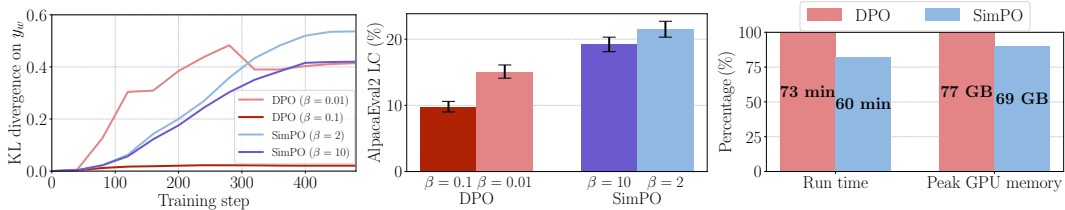

(a) KL divergence w/ different $\beta$. (b) Performance w/ different $\beta$. (c) Efficiency of DPO vs. SimPO.

Figure 5: Comparison between SimPO and DPO (continued). (a) With different $\beta$ in DPO and SimPO, KL divergence from the policy model to the reference model on $y_w$. (b) AlpacaEval2 LC win rate of DPO and SimPO with different $\beta$. (c) Runtime and memory usage for DPO and SimPO.

implicitly counteract length bias. As shown in Table 6 and Figure 4a, employing DPO reduces the Spearman correlation coefficient between average log likelihood and response length compared to the approach without any length normalization (referred to as "SimPO w/o LN"). However, it still exhibits a stronger positive correlation when compared to SimPO.[10]

**DPO reward mismatches generation likelihood.** There is a divergence between DPO's reward formulation, $r_\theta(x, y) = \beta \log \frac{\pi_\theta(y|x)}{\pi_{\text{ref}}(y|x)}$, and the average log likelihood metric, $p_\theta(y \mid x) = \frac{1}{|y|} \log \pi_\theta(y \mid x)$, which directly impacts generation. As shown in Figure 4b, among the instances on the UltraFeedback training set where $r_\theta(x, y_w) > r_\theta(x, y_l)$, almost half of the pairs have $p_\theta(y_w \mid x) < p_\theta(y_l \mid x)$. In contrast, SimPO directly

Table 6: Spearman correlation $\rho$ between average log likelihood of different models and response length on a held-out set.

|  | SimPO w/o LN | DPO | SimPO |
|---|---|---|---|
| $\rho$ | 0.82 | 0.59 | 0.34 |

employs the average log likelihood (scaled by $\beta$) as the reward expression, thereby eliminating the discrepancy completely, as demonstrated in Figure 6b.

**DPO lags behind SimPO in terms of reward accuracy.** In Figure 4c, we compare the reward accuracy of SimPO and DPO, assessing how well their final learned rewards align with preference labels on a held-out set. SimPO consistently achieves higher reward accuracy than DPO, suggesting that our reward design facilitates better generalization and leads to higher quality generations.

**KL divergence of SimPO and DPO.** In Figure 5a, we present the KL divergence between the policy model trained with DPO and SimPO and the reference model with different $\beta$, measured on the winning responses from a held-out set during training. Figure 5b shows the corresponding AlpacaEval 2 LC win rate. Although SimPO does not apply any form of regularization against the reference model, the KL divergence of SimPO is reasonably small. Increasing $\beta$ reduces the KL divergence for both DPO and SimPO, with DPO exhibiting a more pronounced reduction at higher $\beta$ values. In this particular setting (Mistral-base), Figure 5b demonstrates that a smaller $\beta$ can

---

[10]Note that this correlation does not fully reflect the generation length. Despite DPO showing a stronger correlation, the length of its generated responses is comparable to or even slightly shorter than those of the SimPO models. Please find more details in Appendix E.

improve AlpacaEval 2 performance, despite the higher KL divergence.[11] We hypothesize that when the reference model is weak, strictly constraining the policy model to the reference model may not be beneficial. As a caveat, while we did not observe any training collapse or degeneration with proper tuning, in principle, SimPO could potentially lead to reward hacking without explicit regularization against the reference model. In such a scenario, the model might achieve a low loss but degenerate.

**SimPO is more memory and compute-efficient than DPO.** Another benefit of SimPO is its efficiency as it does not use a reference model. Figure 5c illustrates the overall run time and per-GPU peak memory usage of SimPO and DPO in the Llama-3-Base setting using 8×H100 GPUs. Compared to a vanilla DPO implementation,[12] SimPO cuts run time by roughly 20% and reduces GPU memory usage by about 10%, thanks to eliminating forward passes with the reference model.

## 5 Related Work

**Reinforcement learning from human feedback.** RLHF is a technique that aligns large language models with human preferences and values [18, 97, 58, 7]. The classical RLHF pipeline typically comprises three phases: supervised fine-tuning [96, 71, 32, 21, 44, 25, 77, 15, 81], reward model training [31, 56, 16, 52, 35, 46], and policy optimization [65, 4]. Proximal Policy Optimization (PPO) [65] is a widely used algorithm in the third stage of RLHF. The RLHF framework is also widely applied to various applications, such as mitigating toxicity [3, 45, 92], ensuring safety [24], enhancing helpfulness [73, 78], searching and navigating the web [57], and improving model reasoning abilities [34]. Recently, [13] has highlighted challenges across the whole RLHF pipeline from preference data collection to model training. Further research has also demonstrated that RLHF can lead to biased outcomes, such as verbose outputs from the model [28, 66, 80].

**Offline vs. iterative preference optimization.** Given that online preference optimization algorithms are complex and difficult to optimize [95, 64], researchers have been exploring more efficient and simpler alternative offline algorithms. Direct Preference Optimization (DPO) [61] is a notable example. However, the absence of an explicit reward model in DPO limits its ability to sample preference pairs from the optimal policy. To address this, researchers have explored augmenting preference data using a trained SFT policy [91] or a refined SFT policy with rejection sampling [55], enabling the policy to learn from data generated by the optimal policy. Further studies have extended this approach to an iterative training setup, by continuously updating the reference model with the most recent policy model or generating new preference pairs at each iteration [27, 42, 62, 82, 87]. In this work, we focus exclusively on offline settings, avoiding any iterative training processes.

**Preference optimization objectives.** A variety of preference optimization objectives have been proposed besides DPO. Ranking objectives allow for comparisons among more than two instances [26, 54, 67, 86]. Another line of work explores simpler preference optimization objectives that do not rely on a reference model [38, 84], similar to SimPO. [8] proposes a method to jointly optimize instructions and responses, finding it effectively improves DPO. [93] focuses on post-training extrapolation between the SFT and the aligned model to further enhance model performance. In this work, we compare SimPO to a series of offline algorithms, including RRHF [86], SLiC-HF [91], DPO [61], IPO [6], CPO [83], KTO [29], ORPO [38], and R-DPO [60], and find that SimPO can outperform them in both efficiency and performance. Recently, [70] proposed a generalized preference optimization framework unifying different offline algorithms, and SimPO can be seen as a special case.

## 6 Conclusion

In this work, we propose SimPO, a simple and effective preference optimization algorithm that consistently outperforms existing approaches across various training setups. By aligning the reward function with the generation likelihood and introducing a target reward margin, SimPO eliminates the need for a reference model and achieves strong performance without exploiting the length bias. Extensive analysis demonstrates that the key designs in SimPO are crucial and validates the efficiency and effectiveness of SimPO. A detailed discussion of the limitations can be found in Appendix A.

---

[11]We observe that in some settings (*e.g.*, Llama-3-Instruct), a large $\beta$ (*e.g.*, $\beta = 10$) leads to better performance.

[12]DPO can be as memory efficient as SimPO if it were implemented to separate the forward passes of the reference model from the actual preference optimization. However, this implementation is not standard practice.

## Acknowledgments

The authors would like to thank Li Dong, Tianyu Gao, Tanya Goyal, Di Jin, Yuchen Lin, Kaifeng Lyu, Sadhika Malladi, Eric Mitchell, Lewis Tunstall, Haoxiang Wang, Wei Xiong, Zhen Xu, Libing Yang, Zhiyu Zhao, and members of the Princeton NLP group for their valuable feedback and discussions. We thank Niklas Muennighoff for his advice on training and reproducing training KTO models. We thank Haoran Xu for helping verify our CPO runs. Mengzhou Xia is supported by an Apple Scholars in AIML Fellowship. This research is also funded by the National Science Foundation (IIS-2211779) and a Sloan Research Fellowship.

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

## A  Limitations

**More in-depth theoretical analysis.** Despite the empirical success and intuitive motivation of SimPO, a more rigorous theoretical analysis is necessary to fully understand the factors contributing to its effectiveness. Additionally, we introduce an additional hyperparameter, the target reward margin, which requires manual tuning. Future work could explore how to determine the optimal margin automatically and provide a more theoretical understanding of SimPO.

**Safety and honesty.** SimPO is designed to optimize the generation quality of language models by pushing the margin between the average log likelihood of the winning response and the losing response to exceed a target reward margin. However, it does not explicitly consider safety and honesty aspects, which are crucial for real-world applications. Future work should explore integrating safety and honesty constraints into SimPO to ensure that the generated responses are not only high-quality but also safe and honest. The dataset used in this work, UltraFeedback [23], primarily focuses on helpfulness, and future research may consider a more comprehensive study utilizing larger-scale preference datasets [39, 90] and evaluation benchmarks [76] that place a strong emphasis on safety aspects. Nonetheless, we observe that this method consistently achieves high TruthfulQA [53] performance compared to other objectives in Table 9, suggesting its potential for safety alignment.

**Performance drop on math.** We observed that preference optimization algorithms generally decrease downstream task performance, particularly on reasoning-heavy tasks like GSM8k, as shown in Table 9. SimPO occasionally results in performance comparable to or worse than DPO. We hypothesize that this may be related to the choice of training datasets, hyperparameters used for training, or a mismatch of chat templates used for downstream task evaluations. One explanation is that the preference optimization objective may not be effectively increasing the likelihood of preferred sequences despite increasing the reward margin. [59] first observed this phenomenon and point out that this can hinder learning from math preference pairs where changing one token can flip the label (e.g., changing $2 + 2 = 4$ to $2 + 2 = 5$). They propose a simple regularization strategy to add back a reference-model calibrated supervised fine-tuning loss to the preference optimization objective, and effectively mitigate this issue. Future work may consider integrating this regularization strategy into SimPO to improve performance on reasoning-heavy tasks.

## B  Implementation Details

We find that hyperparameter tuning is crucial for achieving optimal performance of preference optimization methods. However, the importance of careful hyperparameter tuning may have been underestimated in prior research, potentially leading to suboptimal baseline results. To ensure a fair comparison, we conduct thorough hyperparameter tuning for all methods compared in our experiments.

**General training hyperparameters.** For the Base training setups, we train SFT models using the UltraChat-200k dataset [25] with the following hyperparameters: a learning rate of 2e-5, a batch size of 128, a max sequence length of 2048, and a cosine learning rate schedule with 10% warmup steps for 1 epoch. All the models are trained with an Adam optimizer [43].

Table 8: The hyperparameter values in SimPO used for each training setting.

| Setting | $\beta$ | $\gamma$ | Learning rate |
|---|---|---|---|
| **Mistral-Base** | 2.0 | 1.6 | 3e-7 |
| **Mistral-Instruct** | 2.5 | 0.3 | 5e-7 |
| **Llama-3-Base** | 2.0 | 1.0 | 6e-7 |
| **Llama-3-Instruct** | 2.5 | 1.4 | 1e-6 |

For the preference optimization stage, we conduct preliminary experiments to search for batch sizes in [32, 64, 128] and training epochs in [1, 2, 3]. We find that a batch size of 128 and a single training epoch generally yield the best results across all methods. Therefore, we fix these values for all preference optimization experiments. Additionally, we set the max sequence length to be 2048 and apply a cosine learning rate schedule with 10% warmup steps on the preference optimization dataset.

**Method-specific training hyperparameters.** We have noticed that the optimal learning rate varies for different preference optimization methods and greatly influences the benchmark performance. Therefore, we individually search the learning rates in the range of [3e-7, 5e-7, 6e-7, 1e-6] for each

Table 7: Various preference optimization objectives and hyperparameter search range.

| Method | Objective | Hyperparameter |
|---|---|---|
| RRHF [86] | $\max\left(0, -\frac{1}{|y_w|}\log\pi_\theta(y_w|x) + \frac{1}{|y_l|}\log\pi_\theta(y_l|x)\right) - \lambda\log\pi_\theta(y_w|x)$ | $\lambda \in [0.1, 0.5, 1.0, 10.0]$ |
| SLiC-HF [91] | $\max\left(0, \delta - \log\pi_\theta(y_w|x) + \log\pi_\theta(y_l|x)\right) - \lambda\log\pi_\theta(y_w|x)$ | $\lambda \in [0.1, 0.5, 1.0, 10.0]$ 
 $\beta \in [0.1, 0.5, 1.0, 2.0]$ |
| DPO [61] | $-\log\sigma\left(\beta\log\frac{\pi_\theta(y_w|x)}{\pi_{\text{ref}}(y_w|x)} - \beta\log\frac{\pi_\theta(y_l|x)}{\pi_{\text{ref}}(y_l|x)}\right)$ | $\beta \in [0.01, 0.05, 0.1]$ |
| IPO [6] | $\left(\log\frac{\pi_\theta(y_w|x)}{\pi_{\text{ref}}(y_w|x)} - \log\frac{\pi_\theta(y_l|x)}{\pi_{\text{ref}}(y_l|x)} - \frac{1}{2\tau}\right)^2$ | $\tau \in [0.01, 0.1, 0.5, 1.0]$ |
| CPO [83] | $-\log\sigma\left(\beta\log\pi_\theta(y_w|x) - \beta\log\pi_\theta(y_l|x)\right) - \lambda\log\pi_\theta(y_w|x)$ | $\lambda = 1.0, \ \beta \in [0.01, 0.05, 0.1]$ |
| KTO [29] | $-\lambda_w\sigma\left(\beta\log\frac{\pi_\theta(y_w|x)}{\pi_{\text{ref}}(y_w|x)} - z_{\text{ref}}\right) + \lambda_l\sigma\left(z_{\text{ref}} - \beta\log\frac{\pi_\theta(y_l|x)}{\pi_{\text{ref}}(y_l|x)}\right),$ 
 where $z_{\text{ref}} = \mathbb{E}_{(x,y)\sim\mathcal{D}}\left[\beta\text{KL}\left(\pi_\theta(y|x)||\pi_{\text{ref}}(y|x)\right)\right]$ | $\lambda_l = \lambda_w = 1.0$ 
 $\beta \in [0.01, 0.05, 0.1]$ |
| ORPO [38] | $-\log p_\theta(y_w|x) - \lambda\log\sigma\left(\log\frac{p_\theta(y_w|x)}{1-p_\theta(y_w|x)} - \log\frac{p_\theta(y_l|x)}{1-p_\theta(y_l|x)}\right),$ 
 where $p_\theta(y|x) = \exp\left(\frac{1}{|y|}\log\pi_\theta(y|x)\right)$ | $\lambda \in [0.1, 0.5, 1.0, 2.0]$ |
| R-DPO [60] | $-\log\sigma\left(\beta\log\frac{\pi_\theta(y_w|x)}{\pi_{\text{ref}}(y_w|x)} - \beta\log\frac{\pi_\theta(y_l|x)}{\pi_{\text{ref}}(y_l|x)} + (\alpha|y_w| - \alpha|y_l|)\right)$ | $\alpha \in [0.05, 0.1, 0.5, 1.0]$ 
 $\beta \in [0.01, 0.05, 0.1]$ |
| SimPO | $-\log\sigma\left(\frac{\beta}{|y_w|}\log\pi_\theta(y_w|x) - \frac{\beta}{|y_l|}\log\pi_\theta(y_l|x) - \gamma\right)$ | $\beta \in [2.0, 2.5]$ 
 $\gamma \in [0.3, 0.5, 1.0, 1.2, 1.4, 1.6]$ |

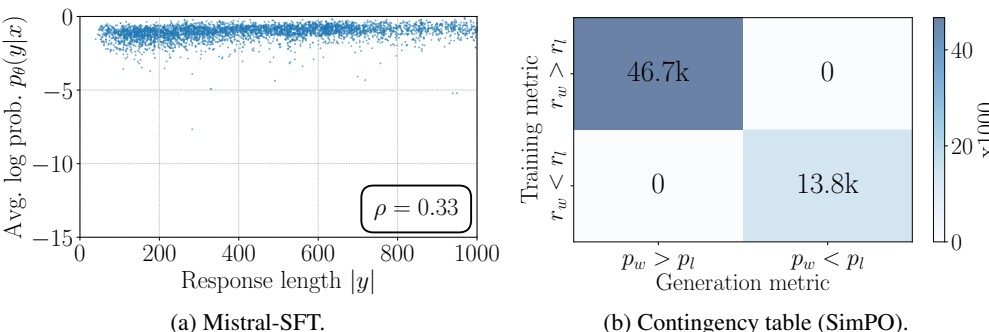

(a) Mistral-SFT.  (b) Contingency table (SimPO).

Figure 6: (a) Likelihood-length correlation plot for Mistral-SFT fine-tuned on UltraChat-200k. (a) Contingency table rankings based on SimPO rewards and the average log likelihood (measured on the training set).

method. Table 7 shows the detailed information on method-specific hyperparameters search ranges for baselines.[13] Table 8 shows SimPO's hyperparameters used under each setting.

**Decoding hyperparameters.** For AlpacaEval 2, we use a sampling decoding strategy to generate responses, with a temperature of 0.7 for the Mistral-Base setting following `zephyr-7b-beta`,[14] a temperature of 0.5 for the Mistral-Instruct setting following `Snorkel-Mistral-PairRM-DPO`, and a temperature of 0.9 for both Llama 3 settings.[15] For Arena-Hard, we use the default greedy decoding for all settings and methods. For MT-Bench, we follow the official decoding configuration which defines different sampling temperatures for different categories.

**Computation environment.** All the training experiments in this paper were conducted on 8×H100 GPUs based on the alignment-handbook repo.[16]

---

[13]There is a discrepancy between the KTO runs in their original paper, where the original runs use a RMSProp optimizer [63]. We use an Adam optimizer [43] for all the experiments.

[14]https://github.com/tatsu-lab/alpaca_eval/blob/main/src/alpaca_eval/models_configs/zephyr-7b-beta/configs.yaml

[15]We grid search the temperature hyperparameter for the Llama-3-Base setting with DPO over $0.1, 0.3, 0.5, 0.7, 0.9$, and fix it for all different methods.

[16]https://github.com/huggingface/alignment-handbook

# C Downstream Task Evaluation

To examine how preference optimization methods affect downstream task performance, we evaluate models trained with different methods on various tasks listed on the Huggingface Open Leaderboard [9]. These tasks include MMLU [36], ARC [19], HellaSwag [89], TruthfulQA [53], Winograd [48], and GSM8K [20]. We follow the established evaluation protocols and present the results for all models in Table 9. Generally, we find that preference optimization's effect varies across tasks.

**Knowledge is largely retained with a small loss.**   Compared to the SFT checkpoint, we find that all preference optimization methods generally maintain MMLU performance with minimal decline. In this aspect, SimPO is largely comparable to DPO.

**Reading comprehension and commonsense reasoning improves.**   For ARC and HellaSwag, preference optimization methods generally improve performance compared to the SFT checkpoint. One hypothesis is that the preference optimization dataset contains similar prompts to these tasks, which helps the model better understand the context and improve reading comprehension and commonsense reasoning abilities.

**Truthfulness improves.**   Surprisingly, we find that preference optimization methods consistently improve TruthfulQA performance compared to the SFT checkpoint, and the improvement could be as high as over 10% in some cases. Similarly, we hypothesize that the preference dataset contains instances that emphasize truthfulness, which helps the model better understand the context and generate more truthful responses.

**Math performance drops.**   GSM8K is the benchmark that shows the most volatility across methods. Notably, except for ORPO, almost all approaches lead to consistent drops in one or more settings. We hypothesize that ORPO retains performance largely due to its supervised fine-tuning loss for regulation. [59] adds a reference-model calibrated supervised fine-tuning loss to the preference optimization objective, and find that it effectively solves the issue and maintains performance on math tasks as well.

Overall, identifying a pattern in downstream performance is challenging. Comprehensive analysis is difficult due to using different pretrained models, preference optimization datasets, and objectives. Recent works indicate that gradient-based approaches could be effective in finding relevant data for downstream tasks [81], and could possibly extended to understand the effect of preference optimization. We believe a thorough study on how preference optimization affects downstream performance would be valuable and call for a rigorous and more comprehensive analysis in future work.

# D Standard Deviation of AlpacaEval 2 and Arena-Hard

We present the standard deviation of AlpacaEval 2 and the 95% confidence interval of Arena-Hard in Table 10. All these metrics are reasonable and do not exhibit any significant outliers or instability.

# E Generation Length Analysis

**Length normalization decreases generation length and improves generation quality.**   Removing length normalization from the SimPO objective results in an approach similar to Contrastive Preference Optimization (CPO) [83], which interpolates reward maximization with a supervised fine-tuning loss and has demonstrated strong performance in machine translation. However, without the supervised fine-tuning loss, the reward maximization objective without length normalization is suboptimal in preference optimization.

We analyze the generation length of models trained with or without length normalization on AlpacaEval 2 and Arena-Hard. As shown in Figure 6, length normalization significantly decrease the generation length by up to 25% compared to when it is not used in most cases. However, even though the generation length is shorter, the models with length normalization consistently achieve much higher win rates on both benchmarks. This suggests that length normalization can effectively control the verbosity of the generated responses, and meanwhile improve the generation quality.

Table 9: Downstream task evaluation results of tasks on the huggingface open leaderboard.

| | MMLU (5) | ARC (25) | HellaSwag (10) | TruthfulQA (0) | Winograd (5) | GSM8K (5) | Average |
|---|---|---|---|---|---|---|---|
| **Mistral-Base** | | | | | | | |
| **SFT** | 60.10 | 58.28 | 80.76 | 40.35 | 76.40 | 28.13 | 57.34 |
| **RRHF** | 57.41 | 52.13 | 80.16 | 43.73 | 76.64 | 4.78 | 52.48 |
| **SLiC-HF** | 59.24 | 55.38 | 81.15 | 48.36 | 77.35 | 33.74 | 59.20 |
| **DPO** | 58.48 | 61.26 | 83.59 | 53.06 | 76.80 | 21.76 | 59.16 |
| **IPO** | 60.23 | 60.84 | 83.30 | 45.44 | 77.58 | 27.14 | 59.09 |
| **CPO** | 59.39 | 57.00 | 80.75 | 47.07 | 76.48 | 33.06 | 58.96 |
| **KTO** | 60.90 | 62.37 | 84.88 | 56.60 | 77.27 | 38.51 | 63.42 |
| **ORPO** | 63.20 | 61.01 | 84.09 | 47.91 | 78.61 | 42.15 | 62.83 |
| **R-DPO** | 59.58 | 61.35 | 84.29 | 46.12 | 76.56 | 18.12 | 57.67 |
| **SimPO** | 59.21 | 62.63 | 83.60 | 50.68 | 77.27 | 22.21 | 59.27 |
| **Mistral-Instruct** | | | | | | | |
| **SFT** | 60.40 | 63.57 | 84.79 | 66.81 | 76.64 | 40.49 | 65.45 |
| **RRHF** | 59.75 | 64.42 | 85.54 | 67.98 | 76.64 | 37.76 | 65.35 |
| **SLiC-HF** | 60.59 | 59.90 | 84.05 | 65.30 | 76.32 | 39.65 | 64.30 |
| **DPO** | 60.53 | 65.36 | 85.86 | 66.71 | 76.80 | 40.33 | 65.93 |
| **IPO** | 60.20 | 63.31 | 84.88 | 67.36 | 75.85 | 39.42 | 65.17 |
| **CPO** | 60.36 | 63.23 | 84.47 | 67.38 | 76.80 | 38.74 | 65.16 |
| **KTO** | 60.52 | 65.78 | 85.49 | 68.45 | 75.93 | 38.82 | 65.83 |
| **ORPO** | 60.43 | 61.43 | 84.32 | 66.33 | 76.80 | 36.85 | 64.36 |
| **R-DPO** | 60.71 | 66.30 | 86.01 | 68.22 | 76.72 | 37.00 | 65.82 |
| **SimPO** | 60.53 | 66.89 | 85.95 | 68.40 | 76.32 | 35.25 | 65.56 |
| **Llama-3-Base** | | | | | | | |
| **SFT** | 64.88 | 60.15 | 81.37 | 45.33 | 75.77 | 46.32 | 62.30 |
| **RRHF** | 64.71 | 62.12 | 82.03 | 55.01 | 77.51 | 44.28 | 64.27 |
| **SLiC-HF** | 64.36 | 61.43 | 81.88 | 54.95 | 77.27 | 48.82 | 64.79 |
| **DPO** | 64.31 | 64.42 | 83.87 | 53.48 | 76.32 | 38.67 | 63.51 |
| **IPO** | 64.40 | 62.88 | 80.46 | 54.20 | 72.22 | 22.67 | 59.47 |
| **CPO** | 64.98 | 61.69 | 82.03 | 54.29 | 76.16 | 46.93 | 64.35 |
| **KTO** | 64.42 | 63.14 | 83.55 | 55.76 | 76.09 | 38.97 | 63.65 |
| **ORPO** | 64.44 | 61.69 | 82.24 | 56.11 | 77.51 | 50.04 | 65.34 |
| **R-DPO** | 64.19 | 64.59 | 83.90 | 53.41 | 75.93 | 39.27 | 63.55 |
| **SimPO** | 64.00 | 65.19 | 83.09 | 59.46 | 77.19 | 31.54 | 63.41 |
| **Llama-3-Instruct** | | | | | | | |
| **SFT** | 67.06 | 61.01 | 78.57 | 51.66 | 74.35 | 68.69 | 66.89 |
| **RRHF** | 67.20 | 61.52 | 79.54 | 53.76 | 74.19 | 66.11 | 67.05 |
| **SLiC-HF** | 66.41 | 61.26 | 78.80 | 53.23 | 76.16 | 66.57 | 67.07 |
| **DPO** | 66.88 | 63.99 | 80.78 | 59.01 | 74.66 | 49.81 | 65.86 |
| **IPO** | 66.52 | 61.95 | 77.90 | 54.64 | 73.09 | 58.23 | 65.39 |
| **CPO** | 67.05 | 62.29 | 78.73 | 54.01 | 73.72 | 67.40 | 67.20 |
| **KTO** | 66.38 | 63.57 | 79.51 | 58.15 | 73.40 | 57.01 | 66.34 |
| **ORPO** | 66.41 | 61.01 | 79.38 | 54.37 | 75.77 | 64.59 | 66.92 |
| **R-DPO** | 66.74 | 64.33 | 80.97 | 60.32 | 74.82 | 43.90 | 65.18 |
| **SimPO** | 65.63 | 62.80 | 78.33 | 60.70 | 73.32 | 50.72 | 65.25 |

**Length is not a reliable indicator of generation quality.** We further analyze the generation length of models trained with different methods on AlpacaEval 2 and Arena-Hard, as shown in Table 10. Generally, we find that no single method consistently generates longer or shorter responses across all settings. Additionally, even though some methods may generate longer responses, they do not necessarily achieve better win rates on the benchmarks. This indicates that the length of the generated responses is not a reliable indicator of generation quality.

**SimPO demonstrates minimal exploitation of response length.** We observe that SimPO has a shorter generation length compared to DPO in the Llama-3-Instruct case but exhibits a higher generation length in other settings, with up to 26% longer responses on AlpacaEval 2. Conversely, SimPO only increases length by only around 5% on Arena-Hard compared to DPO. It is fair to say that the generation length heavily depends on the evaluation benchmark. A stronger indicator is that SimPO consistently achieves a higher length-controlled win rate on AlpacaEval 2 compared to the raw win rate, demonstrating minimal exploitation of response length.

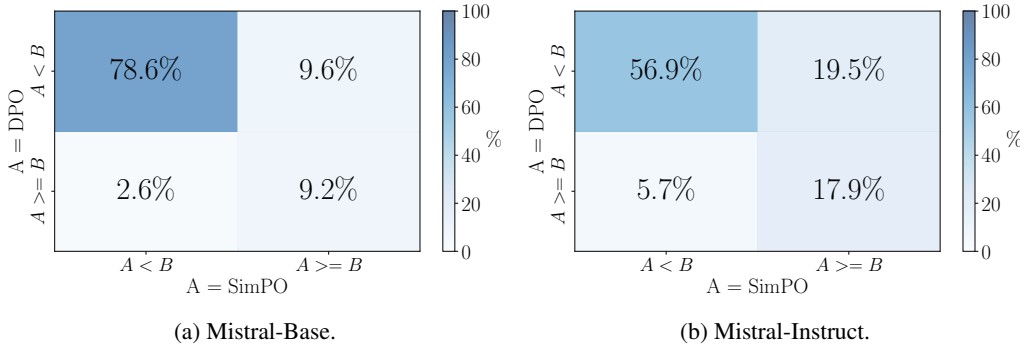

(a) Mistral-Base.

(b) Mistral-Instruct.

Figure 7: Win rate heatmap of Mistral-Base and Mistral-Instruct on AlpacaEval 2. $B$ represents the baseline model (*i.e.*, GPT-4-Preview-1106).

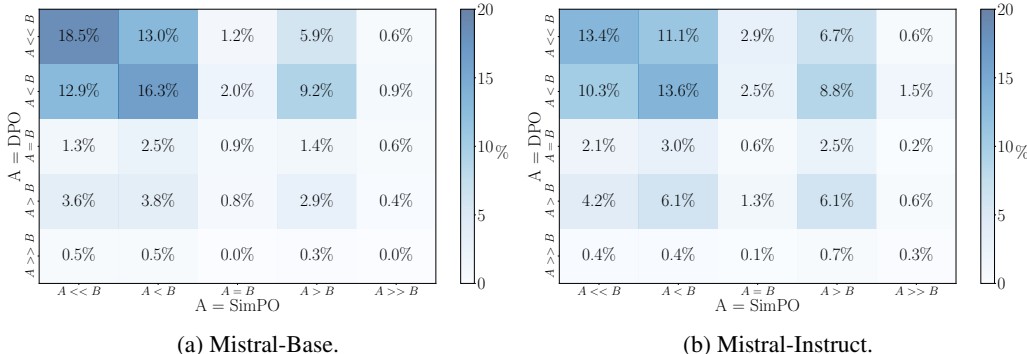

(a) Mistral-Base.

(b) Mistral-Instruct.

Figure 8: Win rate heatmap of Mistral-Base and Mistral-Instruct on Arena-Hard. $B$ represents the baseline model (*i.e.*, GPT-4-0314).

## F Gradient Analysis

We examine the gradients of SimPO and DPO to understand their different impact on the training process.

$$
\nabla_\theta \mathcal{L}_{\text{SimPO}}(\pi_\theta) = -\beta \mathbb{E}_{(x,y_w,y_l)\sim\mathcal{D}} \left[ s_\theta \cdot \left( \underbrace{\frac{1}{|y_w|} \nabla_\theta \log \pi_\theta(y_w|x)}_{\text{increase likelihood on } y_w} - \underbrace{\frac{1}{|y_l|} \nabla_\theta \log \pi_\theta(y_l|x)}_{\text{decrease likelihood on } y_l} \right) \right],
$$

$$
\nabla_\theta \mathcal{L}_{\text{DPO}}(\pi_\theta) = -\beta \mathbb{E}_{(x,y_w,y_l)\sim\mathcal{D}} \left[ d_\theta \cdot \left( \underbrace{\nabla_\theta \log \pi_\theta(y_w|x)}_{\text{increase likelihood on } y_w} - \underbrace{\nabla_\theta \log \pi_\theta(y_l|x)}_{\text{decrease likelihood on } y_l} \right) \right],
$$

(7)

where

$$
s_\theta = \sigma\left( \frac{\beta}{|y_l|} \log \pi_\theta(y_l|x) - \frac{\beta}{|y_w|} \log \pi_\theta(y_w|x) + \gamma \right), \quad d_\theta = \sigma\left( \beta \log \frac{\pi_\theta(y_l|x)}{\pi_{\text{ref}}(y_l|x)} - \beta \log \frac{\pi_\theta(y_w|x)}{\pi_{\text{ref}}(y_w|x)} \right)
$$

represent the gradient weight in SimPO and DPO, respectively. It can be seen that the differences are twofold: (1) comparing the gradient weights $s_\theta$ and $d_\theta$, SimPO's gradient weight $s_\theta$ does not involve the reference model and has a straightforward interpretation: the weights will be higher for samples where the policy model incorrectly assigns higher likelihood to $y_l$ than $y_w$; (2) comparing the gradient updates, SimPO's gradients on $y_l$ and $y_w$ are length-normalized, while DPO's are not. This corresponds to the empirical findings [60] that DPO may exploit length bias: longer sequences with more tokens will receive larger gradient updates in DPO, dominating the training process.

# G    Qualitative Analysis

We present the win rate heatmap of Mistral-Base and Mistral-Instruct on AlpacaEval 2 and Arena-Hard in Figure 7 and Figure 8, respectively. Based on this analysis, we present qualitative examples of responses generated by a SimPO model, a DPO model and the baseline model GPT-4-Preview-1106 on AlpacaEval 2.

**Comparing SimPO with DPO.**    In Figure 9 and Figure 10, we present an example where Mistral-Base-SimPO generates a better-structured answer compared to Mistral-Base-DPO. Given the question, "How can you determine if a person is genuinely interested in a conversation or simply being polite?", the DPO model generates a response with a long list of bullet points, making it difficult to understand the relationships between different points. In contrast, the SimPO model produces a well-structured answer with high-level categorization of different behaviors, followed by detailed suggestions for each category. This makes the answer more readable and easier to understand.

**Comparing Instruct models with Base models when trained with SimPO.**    In Figure 11, we present an example where Llama-3-Instruct generates a more detailed and well-formatted answer compared to the baseline model, and as well as the Llama-3-Base-SimPO model. Given the question: What language does Argentina people speak? Llama-3-Base-SimPO only gives a very brief answer. GPT-4-Preview-1106 gives a more detailed answer in explaining how the Argentina Spanish differs from standard Spanish. However, the answer is not well formatted and a bit hard to parse. Llama-3-Instruct-SimPO gives a detailed and well-formatted answer, which is easier to read and understand, and offers sufficient details.

Figure 9: An AlpacaEval 2 generation from the MistralBase model after training with DPO.

| | |
|---|---|
| **Input** $\rightarrow$ | Q: How can you determine if a person is genuinely interested in a conversation or simply being polite? |
| **Mistral-Base DPO** $\rightarrow$ | As an AI, I dont́ have access to contextual information or nonverbal cues, but here are some general signs that may indicate whether someone is genuinely interested in a conversation or being polite: |

1. Active listening: A genuinely interested person will actively listen to what youŕe saying, ask questions, and provide feedback or comments. They may also maintain eye contact, nod, or use verbal cues like "uh-huh" or "I see" to show theyŕe engaged.

2. Follow-up questions: A genuine interest in the conversation often leads to follow-up questions. If someone is asking questions that show theyŕe interested in learning more about the topic, itś a good sign theyŕe engaged.

3. Body language: Nonverbal cues like leaning in, smiling, and mirroring your body language can indicate genuine interest. However, note that some people may have different body language styles, so this may not always be a reliable indicator.

4. Topic continuation: If the conversation flows naturally, and both parties are contributing equally, itś a sign of mutual interest. If one person dominates the conversation or keeps steering it back to their own interests, it may indicate a lack of genuine interest.

5. Time spent: If the conversation lasts longer than expected or planned, itś a sign that both parties are engaged and interested.

6. Follow-up actions: If someone shows genuine interest, they may follow up with you after the conversation, whether itś through a message, email, or phone call.

Remember, however, that sometimes people may be polite or professional, even if theyŕe not genuinely interested. Itś essential to consider the context and the relationship you have with the person to determine their level of interest accurately.

Table 10: Detailed results of AlpacaEval 2 and Arena-Hard. LC means length-controlled win rate, WR means raw win rate, and STD means standard deviation of win rate. Length is the average generation length. For Arena-Hard, we report the win rate and 95% confidence interval.

| Models | AlpacaEval 2 | | | | Arena-Hard | | | |
|---|---|---|---|---|---|---|---|---|
| | LC (%) | WR (%) | STD (%) | Length | WR | 95 CI high | 95 CI low | Length |
| **Mistral-Base** | | | | | | | | |
| **SFT** | 8.4 | 6.2 | 1.1 | 914 | 1.3 | 1.8 | 0.9 | 521 |
| **RRHF** | 11.6 | 10.2 | 0.9 | 1630 | 6.9 | 8.0 | 6.0 | 596 |
| **SLiC-HF** | 10.9 | 8.9 | 0.9 | 1525 | 7.3 | 8.5 | 6.2 | 683 |
| **DPO** | 15.1 | 12.5 | 1.0 | 1477 | 10.4 | 11.7 | 9.4 | 628 |
| **IPO** | 11.8 | 9.4 | 0.9 | 1380 | 7.5 | 8.5 | 6.5 | 674 |
| **CPO** | 9.8 | 8.9 | 0.9 | 1827 | 5.8 | 6.7 | 4.9 | 823 |
| **KTO** | 13.1 | 9.1 | 0.9 | 1144 | 5.6 | 6.6 | 4.7 | 475 |
| **ORPO** | 14.7 | 12.2 | 1.0 | 1475 | 7.0 | 7.9 | 5.9 | 764 |
| **R-DPO** | 17.4 | 12.8 | 1.0 | 1335 | 9.9 | 11.1 | 8.4 | 528 |
| **SimPO** | 21.4 | 20.8 | 1.2 | 1868 | 16.6 | 18.0 | 15.1 | 699 |
| **Mistral-Instruct** | | | | | | | | |
| **SFT** | 17.1 | 14.7 | 1.1 | 1676 | 12.6 | 14.1 | 11.1 | 486 |
| **RRHF** | 25.3 | 24.8 | 1.3 | 1927 | 18.1 | 19.5 | 16.4 | 517 |
| **SLiC-HF** | 24.1 | 24.6 | 1.3 | 2088 | 18.9 | 20.6 | 17.3 | 578 |
| **DPO** | 26.8 | 24.9 | 1.3 | 1808 | 16.3 | 18.0 | 15.2 | 518 |
| **IPO** | 20.3 | 20.3 | 1.2 | 2024 | 16.2 | 17.9 | 14.4 | 740 |
| **CPO** | 23.8 | 28.8 | 1.3 | 3245 | 22.6 | 25.0 | 20.8 | 812 |
| **KTO** | 24.5 | 23.6 | 1.3 | 1901 | 17.9 | 20.3 | 16.1 | 496 |
| **ORPO** | 24.5 | 24.9 | 1.3 | 2022 | 20.8 | 22.5 | 19.1 | 527 |
| **R-DPO** | 27.3 | 24.5 | 1.3 | 1784 | 16.1 | 18.0 | 14.6 | 495 |
| **SimPO** | 32.1 | 34.8 | 1.4 | 2193 | 21.0 | 22.7 | 18.8 | 539 |
| **Llama-3-Base** | | | | | | | | |
| **SFT** | 6.2 | 4.6 | 0.7 | 1082 | 3.3 | 4.0 | 2.6 | 437 |
| **RRHF** | 10.8 | 8.1 | 0.9 | 1186 | 6.6 | 7.5 | 5.7 | 536 |
| **SLiC-HF** | 12.1 | 10.1 | 0.9 | 1540 | 10.3 | 11.5 | 8.9 | 676 |
| **DPO** | 18.2 | 15.5 | 1.1 | 1585 | 15.9 | 18.1 | 14.1 | 563 |
| **IPO** | 14.4 | 14.2 | 1.1 | 1856 | 17.8 | 19.5 | 16.0 | 608 |
| **CPO** | 12.3 | 13.7 | 1.0 | 2495 | 11.6 | 13.2 | 10.4 | 800 |
| **KTO** | 14.2 | 12.4 | 1.0 | 1646 | 12.5 | 14.2 | 10.9 | 519 |
| **ORPO** | 12.2 | 10.6 | 0.9 | 1628 | 10.8 | 12.3 | 9.6 | 639 |
| **R-DPO** | 17.6 | 14.4 | 1.1 | 1529 | 17.2 | 18.5 | 15.7 | 527 |
| **SimPO** | 22.0 | 20.3 | 1.2 | 1795 | 23.4 | 25.4 | 21.6 | 704 |
| **Llama-3-Instrct** | | | | | | | | |
| **SFT** | 26.0 | 25.3 | 1.3 | 1920 | 22.3 | 23.9 | 20.3 | 596 |
| **RRHF** | 31.3 | 28.4 | 1.33 | 1805 | 26.5 | 28.4 | 24.6 | 502 |
| **SLiC-HF** | 26.9 | 27.5 | 1.3 | 1977 | 26.2 | 28.4 | 24.4 | 584 |
| **DPO** | 40.3 | 37.9 | 1.4 | 1883 | 32.6 | 34.8 | 30.3 | 528 |
| **IPO** | 35.6 | 35.6 | 1.4 | 1983 | 30.5 | 32.8 | 28.4 | 554 |
| **CPO** | 28.9 | 32.2 | 1.4 | 2166 | 28.8 | 30.6 | 26.6 | 624 |
| **KTO** | 33.1 | 31.8 | 1.4 | 1909 | 26.4 | 28.7 | 24.3 | 536 |
| **ORPO** | 28.5 | 27.4 | 1.3 | 1888 | 25.8 | 27.4 | 23.8 | 535 |
| **R-DPO** | 41.1 | 37.8 | 1.4 | 1854 | 33.1 | 35.3 | 30.9 | 522 |
| **SimPO** | 44.7 | 40.5 | 1.4 | 1825 | 33.8 | 35.9 | 32.0 | 504 |

Table 11: Average response lengths on AlpacaEval 2 and Arena-Hard trained with Mistral-Base or Mistral-Instruct.

| Model | AlpacaEval 2 | | | | Arena-Hard | | | |
|---|---|---|---|---|---|---|---|---|
| | Mistral-Base | | Mistral-Instruct | | Mistral-Base | | Mistral-Instruct | |
| | LC (%) | Length | LC (%) | Length | WR (%) | Length | WR (%) | Length |
| **SimPO** | 21.5 | 1868 | 32.1 | 2193 | 16.6 | 699 | 21.0 | 539 |
| **SimPO w/o LN** | 11.9 | 2345 | 19.1 | 2067 | 9.4 | 851 | 16.3 | 679 |

Figure 10: An AlpacaEval 2 generation from the Mistral-Base model after training with SimPO. Compared to the output generated by the DPO model, as shown in Figure 9, the generation by SimPO is better structured with hierarchical discussions, making the information more clearly presented and readable.

| | |
|---|---|
| **Input** → | Q: How can you determine if a person is genuinely interested in a conversation or simply being polite? |
| **Mistral-Base SimPO** → | Here are some indicators to consider:

1. Engagement:
– Active listening: Does the person maintain eye contact, nod, ask follow-up questions, and respond thoughtfully to what you say? Engaged individuals tend to show genuine interest by demonstrating attentiveness and participation in the conversation.
– Initiative: Does the person take the lead in steering the conversation, expressing enthusiasm, sharing personal experiences, or asking questions beyond polite small talk? Someone who genuinely cares about the topic or connection may show initiative in keeping the discussion engaging.
– Body language: Observe nonverbal cues such as leaning in, mirroring posture, facial expressions, and gestures that convey enthusiasm, curiosity, or openness. These signals suggest a deeper level of interest beyond mere politeness.

2. Content and depth:
– Topic selection: Does the person show genuine curiosity or passion about the subject matter, asking relevant questions, sharing insights, or expressing opinions? If they seem invested in exploring the topic's nuances, it indicates a genuine interest beyond mere politeness.
– Level of detail: Does the person provide specific examples, personal anecdotes, or elaborate on their thoughts, indicating a deeper level of engagement and investment in the conversation?

3. Follow-up actions:
– Continued communication: Does the person initiate further conversations, send follow-up messages, or suggest meeting up outside the initial interaction? Such actions suggest a genuine desire to maintain the connection and explore further topics, indicating a genuine interest beyond mere politeness.
– Personalization: Does the person remember details discussed earlier, reference shared interests, or show interest in learning more about your life or experiences? Personalized interactions indicate a genuine interest in building a relationship beyond mere politeness.

Overall, genuine interest in a conversation involves active participation, enthusiasm, depth, and follow-up actions that demonstrate a genuine desire to connect and engage beyond mere politeness. |

Figure 11: A case study on AlpacaEval 2 demonstrates that Llama-3-Instruct, trained with SimPO, provides a better formatted and more detailed answer than both Llama-3-Base, also trained with SimPO, and the baseline model GPT-4-1106-Preview. This illustrates how the instruction setting typically outperforms the base setting.

| | |
|---|---|
| **Input** → | Q: what language does argentina people speak |
| **Llama-3-Base SimPO** → | The official language spoken in Argentina is Spanish, specifically Argentine Spanish, which shares similarities with other varieties of Spanish spoken in neighboring countries like Uruguay, Chile, and Paraguay. |
| **Llama-3-Instruct SimPO** → | In Argentina, the official language is Spanish. Specifically, the dialect spoken in Argentina is known as Rioplatense Spanish, which is a variation of Spanish with some unique characteristics and influences from Italian, German, and other immigrant languages.

Rioplatense Spanish is distinguished by:
1. **Voseo:** Instead of using *tú* (you) as the informal singular pronoun, Argentines use *vos* and its associated verb conjugations.
2. **Lunfardo:** A slang vocabulary derived from Italian, Corsican, and other immigrant languages, which has been incorporated into everyday speech.
3. **Accent and pronunciation:** Argentines tend to pronounce the "c" and "z" sounds with a soft "s" or "sh" sound, rather than a hard "th" sound like in standard Spanish.

Overall, Argentine Spanish is known for its distinctive flavor and is widely understood throughout the Spanish-speaking world. However, it's worth noting that there may be regional variations within Argentina, and some dialects may differ slightly from the standard Rioplatense variety. |
| **gpt4-turbo-1106-preview** → | The primary language spoken in Argentina is Spanish. Specifically, the variety of Spanish spoken there is known as Rioplatense Spanish, which is characterized by its unique accent and the use of the pronoun "vos" instead of "tú" for the second person singular. Additionally, there are various indigenous languages spoken in different regions of the country, as well as immigrant languages such as Italian, German, and English, but these are much less widespread than Spanish. |

# H   Llama-3-Instruct v0.2 (Jul 7, 2024)

Table 12: Results of Llama-3-Instruct (8B) setting, utilizing preference labels annotated by a stronger reward model (ArmoRM [79], we term it as version 0.2).

| Method | Llama-3-Instruct (8B) | | | | | | |
| | AlpacaEval 2 | | | Arena-Hard | | MT-Bench | |
| | LC (%) | WR (%) | Length | WR (%) | Length | GPT-4 Turbo | GPT-4 |
|---|---|---|---|---|---|---|---|
| SFT | 26.0 | 25.3 | 1920 | 22.3 | 596 | 6.9 | 8.1 |
| SimPO v0.1 | 44.7 | 40.5 | 1825 | 33.8 | 504 | 7.0 | 8.0 |
| RRHF [86] | 37.9 | 31.6 | 1700 | 28.8 | 467 | 7.1 | 8.2 |
| SLiC-HF [91] | 33.9 | 32.5 | 1938 | 29.3 | 599 | 6.9 | 8.1 |
| DPO [61] | 48.2 | **47.5** | 2000 | 35.2 | 609 | 7.0 | 8.2 |
| IPO [6] | 46.8 | 42.4 | 1830 | **36.6** | 527 | **7.2** | 8.2 |
| CPO [83] | 34.1 | 36.4 | 2086 | 30.9 | 604 | **7.2** | 8.2 |
| KTO [29] | 34.1 | 32.1 | 1878 | 27.3 | 541 | **7.2** | 8.2 |
| ORPO [38] | 38.1 | 33.8 | 1803 | 28.2 | 520 | **7.2** | **8.3** |
| R-DPO [60] | 48.0 | 45.8 | 1933 | 35.1 | 608 | 7.0 | 8.2 |
| SimPO v0.2 | **53.7** | **47.5** | 1777 | 36.5 | 530 | 7.0 | 8.0 |

Table 13: Downstream task evaluation results of tasks on the huggingface open leaderboard.

| | MMLU (5) | ARC (25) | HellaSwag (10) | TruthfulQA (0) | Winograd (5) | GSM8K (5) | Average |
|---|---|---|---|---|---|---|---|
| | | | | Llama-3-Instruct | | | |
| **SFT** | 67.06 | 61.01 | 78.57 | 51.66 | 74.35 | 68.69 | 66.89 |
| **RRHF** | 67.20 | 61.52 | 79.54 | 53.76 | 74.19 | 66.11 | 67.05 |
| **SLiC-HF** | 66.41 | 61.26 | 78.80 | 53.23 | 76.16 | 66.57 | 67.07 |
| **DPO** | 66.88 | 63.99 | 80.78 | 59.01 | 74.66 | 49.51 | 65.86 |
| **IPO** | 66.52 | 61.95 | 77.90 | 54.64 | 73.09 | 58.23 | 65.39 |
| **CPO** | 67.05 | 62.29 | 78.73 | 54.01 | 73.72 | 67.40 | 67.20 |
| **KTO** | 66.38 | 63.57 | 79.51 | 58.15 | 73.40 | 57.01 | 66.34 |
| **ORPO** | 66.41 | 61.01 | 79.38 | 54.37 | 75.77 | 64.59 | 66.92 |
| **R-DPO** | 66.74 | 64.33 | 80.97 | 60.32 | 74.82 | 43.90 | 65.18 |
| **SimPO** | 65.63 | 62.80 | 78.33 | 60.70 | 73.32 | 50.72 | 65.25 |
| | | | | Llama-3-Instruct v0.2 | | | |
| **SFT** | 67.06 | 61.01 | 78.57 | 51.66 | 74.35 | 68.69 | 66.89 |
| **RRHF** | 66.60 | 63.74 | 80.98 | 59.40 | 76.32 | 58.68 | 67.62 |
| **SLiC-HF** | 66.91 | 61.77 | 79.17 | 56.36 | 76.40 | 68.23 | 68.14 |
| **DPO** | 67.33 | 64.08 | 80.08 | 56.33 | 75.61 | 54.51 | 66.32 |
| **IPO** | 67.32 | 63.23 | 78.71 | 58.12 | 74.51 | 56.33 | 66.37 |
| **CPO** | 66.86 | 62.80 | 79.10 | 55.62 | 73.88 | 67.78 | 67.67 |
| **KTO** | 67.25 | 63.57 | 79.66 | 55.56 | 74.98 | 66.41 | 67.91 |
| **ORPO** | 66.78 | 63.40 | 80.09 | 57.52 | 76.72 | 66.72 | 68.54 |
| **R-DPO** | 67.28 | 64.51 | 80.22 | 56.44 | 75.61 | 52.99 | 66.17 |
| **SimPO** | 66.51 | 66.64 | 78.97 | 63.86 | 74.74 | 55.65 | 67.73 |
| **SimPO w/ SFT** | 66.74 | 63.82 | 78.82 | 60.52 | 73.72 | 64.06 | 67.95 |

In this section, we update the Llama-3-Instruct setting, primarily by utilizing a stronger reward model to annotate our generated preference dataset.

**Enhanced reward model yields significantly better results.**   In our previous version, we use PairRM [41] as our reward model to rank generated candidate responses. The results, presented in Table 12, show that switching the reward model from PairRM [41] to ArmoRM [79] for ranking the data markedly improves model performance. This underscores the importance of a high-quality preference optimization dataset for enhancing performance. Notably, SimPO has achieved a 53.7 LC win rate on AlpacaEval 2 and 36.5 on Arena-Hard, surpassing the previous version by 9.0 and 2.7 points, respectively.

We use the following hyperparameters for SimPO under the Llama-3 Instruct v0.2 setting: $\beta = 10$ and $\gamma = 3$. The other hyperparameters (*e.g.*, learning rate, batch size, max sequence lengths) are kept the same as the original Llama-3-8B-Instruct setting.

**Strong SFT model and high-quality policy data diminish algorithm differences.** With a strong SFT model like Llama-3-8B-Instruct, and as the preference optimization data quality improves, the differences between algorithms become less pronounced. For instance, DPO achieved a similar win rate as SimPO in terms of raw win rate, and DPO, IPO, and R-DPO all exhibited comparable raw win rates on Arena-Hard. However, SimPO maintains an advantage by producing shorter sequences, resulting in a significantly better LC win rate on AlpacaEval 2.

**Stronger downstream task performance.** The v0.2 version also shows improved performance in downstream tasks across various objectives. However, DPO, IPO, R-DPO, and SimPO continue to experience a decline in reasoning-intensive domains such as GSM8K. In contrast, objectives that include an SFT component maintain their performance in mathematical tasks.

Table 14: AlpacaEval 2 performance of SimPO and SimPO with an SFT loss.

| Method | LC (%) | WR (%) |
|---|---|---|
| SimPO v0.2 | 53.7 | 47.5 |
| w/ SFT | 41.4 | 36.5 |

**Incorporating SFT regularization in SimPO.** Several reference-free algorithms, including RRHF [86], SLiC-HF [91], CPO [83], and ORPO [38], employ SFT regularization in their objectives. SFT regularization can be an effective method to prevent reward hacking, ensuring that the solution maintains low loss without resulting in degraded generations. We also experiment with the integration of an SFT loss in SimPO, yielding the following objective:

$$\mathcal{L}_{\text{SimPO w/ SFT}}(\pi_\theta) = -\mathbb{E}_{(x,y_w,y_l)\sim\mathcal{D}}\left[\log\sigma\left(\frac{\beta}{|y_w|}\log\pi_\theta(y_w|x) - \frac{\beta}{|y_l|}\log\pi_\theta(y_l|x) - \gamma\right) + \lambda\log\pi_\theta(y_w|x)\right].$$

As shown in Table 14, the addition of the SFT regularization leads to a decrease in performance on AlpacaEval 2. However, we note that SFT regularization provides substantial benefits to certain tasks such as GSM8K, as shown in Table 12. These contrasting results suggest that the impact of SFT in preference optimization may vary depending on the training setup and the nature of the task. Further comprehensive studies on this topic are left for future research.

## I Applying Length Normalization and Target Reward Margin to DPO (Jul 7, 2024)

Since the release of the paper, we have had inquiries from researchers about whether the key design elements of SimPO—length normalization and target reward margin—could benefit DPO. By doing so, we will derive the following two objectives:

$$\mathcal{L}_{\text{DPO w/ LN}}(\pi_\theta; \pi_{\text{ref}}) = -\mathbb{E}_{(x,y_w,y_l)\sim\mathcal{D}}\left[\log\sigma\left(\frac{\beta}{|y_w|}\log\frac{\pi_\theta(y_w\mid x)}{\pi_{\text{ref}}(y_w\mid x)} - \frac{\beta}{|y_l|}\log\frac{\pi_\theta(y_l\mid x)}{\pi_{\text{ref}}(y_l\mid x)}\right)\right].$$

$$\mathcal{L}_{\text{DPO w/ }\gamma}(\pi_\theta; \pi_{\text{ref}}) = -\mathbb{E}_{(x,y_w,y_l)\sim\mathcal{D}}\left[\log\sigma\left(\beta\log\frac{\pi_\theta(y_w\mid x)}{\pi_{\text{ref}}(y_w\mid x)} - \beta\log\frac{\pi_\theta(y_l\mid x)}{\pi_{\text{ref}}(y_l\mid x)} - \gamma\right)\right].$$

An intuitive understanding of how length normalization could benefit DPO is that, despite DPO's reward design being implicitly normalized by the reference model, the policy model might still exploit length bias from the data, resulting in a disproportionately high probability for longer sequences. Applying length normalization could help mitigate this effect.

We train models with the objectives mentioned above and compare their performance to that of DPO and SimPO, as shown in Table 15.

The results indicate that, unlike SimPO, length normalization and target reward margin do not consistently benefit DPO. Specifically, length normalization significantly improves DPO performance only in the Mistral-Base setting, where the preference optimization dataset shows a strong length bias. However, it does not provide a benefit in the Mistral-Instruct setting, where the lengths of winning and losing responses are comparable. This is likely because DPO already includes an implicit instance-wise target reward margin via the reference model, as shown in the derivation below.

$$\mathcal{L}_{\text{DPO}} = \log\sigma\left(\beta\log\frac{\pi_\theta(y_w\mid x)}{\pi_{\text{ref}}(y_w\mid x)} - \beta\log\frac{\pi_\theta(y_l\mid x)}{\pi_{\text{ref}}(y_l\mid x)}\right)$$

$$= \log\sigma\left(\beta\log\pi_\theta(y_w\mid x) - \beta\log\pi_\theta(y_l\mid x) - \underbrace{(\beta\log\pi_{\text{ref}}(y_w\mid x) - \beta\log\pi_{\text{ref}}(y_l\mid x))}_{=\gamma_{\text{ref}}}\right).$$

Table 15: Applying length normalization (LN) and target reward margin ($\gamma$) to DPO.

| Method | Mistral-Base (7B) Setting | | | | | Mistral-Instruct (7B) Setting | | | | |
| | AlpacaEval 2 | | Arena-Hard | MT-Bench | | AlpacaEval 2 | | Arena-Hard | MT-Bench | |
| | LC (%) | WR (%) | WR (%) | GPT-4 Turbo | GPT-4 | LC (%) | WR (%) | WR (%) | GPT-4 Turbo | GPT-4 |
|---|---|---|---|---|---|---|---|---|---|---|
| SimPO | 21.5 | 20.8 | 16.6 | 6.0 | 7.3 | 32.1 | 34.8 | 21.0 | 6.6 | 7.6 |
| DPO | 15.1 | 12.5 | 10.4 | 5.9 | 7.3 | 26.8 | 24.9 | 16.3 | 6.3 | 7.6 |
| w/ LN | 21.0 | 17.7 | 15.2 | 5.9 | 7.2 | 21.7 | 20.9 | 15.6 | 6.4 | 7.7 |
| w/ $\gamma$ | 15.2 | 12.1 | 10.3 | 5.7 | 7.3 | 23.0 | 24.6 | 14.7 | 6.3 | 7.6 |

## J   Applying SimPO to Gemma 2 Models (Sept 16, 2024)

**Performance degradation on other benchmarks for Llama-3-SimPO checkpoints.**   After releasing the Llama-3-SimPO checkpoints, we received extensive feedback about performance degradation on benchmarks measuring specific capabilities, such as MMLU and GSM8K. To investigate this issue, we continued training the Llama-3-8B-Instruct model with different learning rates, as reported in Table 16. We find that using a higher learning rate results in a stronger model in chat-oriented benchmarks, at the cost of catastrophic forgetting on GSM8K and MMLU.[17] With a smaller learning rate, the model's performance on chat benchmarks is slightly worse, but its performance on GSM8K and MMLU is better retained. This demonstrates a trade-off between chat-oriented benchmarks and other benchmarks when continuing training from a strong instruction-tuned model.

Table 16: Results on AlpacaEval 2, ZeroEval GSM, and ZeroEval MMLU when continuing training from Llama-3-8B-Instruct with different learning rates. * indicates the released checkpoint.

| Model | AlpacaEval 2 LC (%) | ZeroEval GSM (%) | ZeroEval MMLU (%) |
|---|---|---|---|
| Llama-3-8B-Instruct | 26.0 | 78.5 | 61.7 |
| SimPO (lr=4e-7) | 38.8 | 77.9 | 62.6 |
| SimPO (lr=5e-7) | 44.6 | 77.0 | 62.3 |
| SimPO (lr=1e-6)* | 53.7 | 57.4 | 54.9 |

**Applying SimPO to Gemma 2 models presents a different trend.**   We evaluate SimPO using Google's recently released Gemma-2-9B-it model [72], which represents a strong open-source model. For training data, we generate up to 5 responses per prompt from the UltraFeedback dataset [23] and use the ArmoRM model [79] to annotate preferences between responses. We compare our SimPO against a DPO-trained variant, both fine-tuned from the Gemma-2-9B-it base model. As shown in Appendix J, SimPO demonstrates superior performance on chat benchmarks like AlpacaEval 2 and Arena-Hard while maintaining the model's original zero-shot capabilities on tasks like GSM8K and MMLU. Notably, we find that varying the learning rate during fine-tuning has minimal impact on the model's performance. These results suggest an underlying property difference between the Llama-3 checkpoints and the Gemma 2 checkpoints, and might be worth further investigation.

**Gemma-2-9B-it-SimPO significantly improved the ranking of the Gemma-2-9B-it model on Chatbot Arena.**   During the development stage, we relied solely on automated metrics to evaluate the model's performance. To determine if these metrics aligned with real user preferences, we submitted our best-performing model, Gemma-2-9B-it-SimPO, to the Chatbot Arena leaderboard hosted by LMSYS [17]. We find that our model improved the original Gemma-2-9B-it ranking from 36th to 25th, making the SimPO variant the top-ranked <10B model on the Chatbot Arena leaderboard based on real user votes as of September 16th, 2024.

---

[17]We evaluate the zero-shot performance of the models on GSM8K and MMLU using the ZeroEval repository which adopts a unified setup.

Table 17: Benchmark performance of Gemma-2-9B trained with DPO and SimPO on UltraFeedback (responses regenerated with Gemma-2-9B-it, following the same dataset construction process as Llama-3-Instruct (8B) described in Section 3). SimPO results in better instruction following performance than DPO without degrading math abilities (GSM) or general knowledge (MMLU) of the original model. * indicates the released checkpoint.

| Model | Instruction Following | | Capabilities | |
| --- | --- | --- | --- | --- |
| | AlpacaEval 2 LC (%) | Arena-Hard (%) | ZeroEval GSM (%) | ZeroEval MMLU (%) |
| Gemma-2-9B-it | 51.1 | 40.8 | 87.4 | **72.7** |
| Gemma-2-9B-DPO | 67.8 | 58.9 | **88.5** | 72.2 |
| Gemma-2-9B-SimPO (lr=6e-7) | 71.7 | 58.3 | 88.3 | 72.2 |
| Gemma-2-9B-SimPO (lr=8e-7)* | **72.4** | **59.1** | 88.0 | 72.2 |
| Gemma-2-9B-SimPO (lr=1e-6) | 71.0 | 58.3 | 87.4 | 71.5 |

