# OpenReview forum: "SimPO: Simple Preference Optimization with a Reference-Free Reward"
_NeurIPS.cc/2024/Conference — NeurIPS 2024 poster_

### Official Review · Reviewer_Wuuc · 2024-07-01

**Soundness:** 3
**Presentation:** 4
**Contribution:** 3
**Rating:** 7
**Confidence:** 3

**Summary:**

This paper presents SimPO, an offline preference optimization method for LLM alignment. SimPO replaces the KL term in DPO with a length-regularized log-probability and adds a margin value for regularization. Extensive experiments in chat benchmarks show that SimPO significantly outperforms DPO and other preference optimization variants, despite its simplicity. The authors conclude that the success of SimPO can be attributed to the better alignment between training and decoding objectives, as well as reduced exploitation of generating lengthy responses.

**Strengths:**

+ This paper is well-written and easy to follow. The motivation is clear, and the main approach is quite understandable.

+ The experiments are extensive and well-designed. In addition to benchmark numbers, the authors also provide an in-depth analysis of generation lengths, log probabilities, and differences in rewards for DPO and SimPO models. This greatly helps readers understand the source of the performance gains.

**Weaknesses:**

I do not find any significant weaknesses beyond the limitations proposed by the authors. Since I generally agree with these limitations, I am reiterating them in my review:

+ A theoretical understanding of SimPO is lacking. Although SimPO's design is not theoretically grounded, it performs well in practice, as validated by the authors' experiments.

+ The experiments in the paper solely focus on evaluating helpfulness, disregarding safety, honesty, etc. I think this is the major weakness. SimPO removes the KL regularization from the reference model, so it might intuitively suffer from safety issues. This problem should be further studied in the paper.

**Questions:**

+ Could the authors explain why length-controlled WR is important? From my perspective, controlling response length is helpful during training because it encourages the model to generate short but useful responses. However, forcing the generation of short responses during evaluation is not necessary. Good responses are good regardless of length. Evaluating length-controlled WR seems to set a privileged metric for SimPO.

+ Since SimPO removes the KL regularization, how does it perform with poor data quality, which is a common application scenario? Is it more fragile and sensitive to data quality?

**Limitations:**

The limitations have been addressed by the authors in Section 6 and in my weaknesses section.

---

> ### Author Rebuttal · Authors · 2024-08-07
>
> We thank the reviewer for acknowledging the clarity and comprehensive evaluation of our paper. We address your raised points as follows.
>
> **KL regularization and safety**:
> We would like to present our most recent results based on the gemma-2-9b-it model. We measured the original gemma-2-9b-it model, its DPO variant, and the SimPO variant using Sorry-Bench (Xie et al., 2024 [1]), a benchmark for testing the refusal behavior of language models when faced with unsafe queries. Our findings indicate that further tuning the Gemma model on the UltraFeedback dataset with either DPO or SimPO increases the attack success rate, probably due to the lack of safety-related data in UltraFeedback. However, we also find that SimPO is safer than DPO. Therefore, we believe that SimPO does not raise more safety concerns than DPO.
>
> |  model  | Sorry-Bench attack success rate |
> |----------------------|:------------------------------:|
> | gemma-2-9b-it        |    10.89% |
> | w/ DPO               |  50.44% |
> | w/ SimPO (without safety preference data)   |  33.33% |
>
> We further direct the reviewer to the PDF attached to our general response, where we demonstrate that even without explicit KL regularization, SimPO achieves a comparable KL divergence from the initial SFT model as KL-regularized objectives like DPO.
>
> [1] [SORRY-Bench: Systematically Evaluating Large Language Model Safety Refusal Behaviors](https://arxiv.org/abs/2406.14598)
>
> **Why is length-controlled WR important**:
> The length-controlled win rate (LC-WR), introduced by Dubois et al. (2024) [2], effectively addresses the length bias issue prevalent in model-based evaluations such as AlpacaEval. This metric has demonstrated a better correlation with human judgments compared to raw win rates [2]. Hence, it has been adopted as the preferred metric for ranking models on the AlpacaEval leaderboard. Given its significance and widespread acceptance in the field, we have incorporated the length-controlled WR as one of our primary evaluation metrics in this study.
>
> [2] [Length-Controlled AlpacaEval: A Simple Way to Debias Automatic Evaluators](https://arxiv.org/abs/2404.04475v1)
>
> **Is SimPO more fragile and sensitive to data quality?**
> Our extensive experiments across various settings do not indicate that SimPO is more susceptible to low-quality data than other methods. In fact, the UltraFeedback dataset used in our study inherently includes a variety of data quality levels, as the responses are generated by randomly-sampled models from a diverse pool, including relatively weak models (e.g., Alpaca-7B). Despite this variability in data quality, SimPO consistently demonstrates significant empirical advantages over other baselines.
>
> We hypothesize that SimPO's robustness to data quality issues stems from two key factors:
> 1. The use of a small learning rate, which is the common practice in preference optimization algorithms and naturally prevents significant divergence from the initial model.
> 2. The inherent robustness of some large language models (e.g., gemma models), which allows them to maintain performance even when exposed to some degree of low-quality input.
>
> These factors combined may provide sufficient protection against catastrophic forgetting, even without the need for explicit KL regularization.

---

> > ### Comment · Reviewer_Wuuc · 2024-08-09
> > **Thank you for your response.**
> >
> > I'v read the authors' comments to all reviewers. I don't have any additional concerns and would like to remain my score, voting for acceptance.

---

> ### Author Response · Authors · 2024-08-12
> **Thank you for the support!**
>
> Dear Reviewer Wuuc,
>
> Thank you once again for your thoughtful review and valuable feedback! We appreciate your support!
>
> Sincerely,
> Authors

---

### Official Review · Reviewer_TXhi · 2024-07-04

**Soundness:** 4
**Presentation:** 4
**Contribution:** 3
**Rating:** 6
**Confidence:** 5

**Summary:**

This paper proposes a new offline-RLHF algorithm SimPO, which significantly improves current DPO variants on a collection of benchmarks.

**Strengths:**

The SimPO algorithm is intuitive, simple to implement, and works well, with a good presentation.

A few concrete strong points are list below.

1. The algorithm is intuitive and the overall design is novel in the current literature. The paper proposes two simple techniques, including directly optimizing the generation probability and adding a discrepancy term. Both two techniques are intuitive. Overall, little efforts in offline RLHF methods have fully explored these two directions systematically.

2. The experiments are solid. This paper conducts experiments on a diverse collection of chat benchmarks. The results are significant.

3. The overall presentation is great and easy to follow.

**Weaknesses:**

The following comments are for potential improvement of this work and future research directions. These are not complaints.

1. Some fine-grained case studies can be beneficial. The paper primarily focuses on quantitative evaluations. But it would be interesting to see concrete changes between DPO and SimPO, i.e., some representative responses generated by SimPO and DPO, to see what has changed. The LN experiments are great. But things can be better.

2. With the current formulation, the SimPO objective reminds me of those online RLHF methods. Assuming a -1/1 reward function, many online RLHF methods optimize similar objectives, i.e., direction optimizing the responses with a 1 reward response (winning responses) while reducing the reward of bad ones (losing responses). It will be interesting to see some discussions between SimPO and online RLHF methods. Also, many recent practices have observed that online DPO works greatly. It would be interesting to see if the same phenomena can be observed for SimPO, which looks much closer to online RLHF variants. This is a potential direction to make this paper a much stronger one.

3. Another possible direction to explore is about tasks other than chat, e.g., reasoning tasks like coding and maths. The evaluations in this paper are indeed sufficient as they are, but some discussions on a broad domain can be appreciated.

**Questions:**

I do not have any specific questions. As a final comment, if possible, I personally like to see some concrete examples of responses generated by SimPO and DPO (beyond the response lengths) to have a better understanding of the actual improvement.

**Limitations:**

Most limitations have been declared by the authors.

---

> ### Author Rebuttal · Authors · 2024-08-07
>
> We thank the reviewer for acknowledging the novelty, simplicity, and significant empirical results of SimPO!
>
> **Fine-grained case studies**:
> Thank you for the suggestion! We refer the reviewer to Figures 8 and 9 in our original submission, as well as Figure 2 in the PDF attached to our general response. These examples intuitively demonstrate why SimPO outperforms DPO:
> * SimPO can produce better-structured answers than DPO. (Figures 8 and 9 in our paper)
> * SimPO can generate more concise and clear responses than DPO. (Figure 2 in the rebuttal PDF)
>
> We acknowledge that qualitative analysis of model outputs is becoming increasingly challenging, given the breadth and depth of the questions in these benchmarks. We will provide more qualitative cases in our next revisions.
>
> **Online methods**:
> Thank you for the suggestions! We are actively working on an online version of SimPO. Specifically, we alternate between generating online preference optimization data and training the model with the generated data. Here are some initial results we have for a stronger version of the model — gemma-2-9B-it:
>
> |  model | LC Win rate | Raw Win Rate |
> |----------------|:-----------:|:------------:|
> | gemma-2-9b-it  |     51.1    |     38.1     |
> | SimPO (1 iter) |     74.6    |     66.7     |
> | SimPO (2 iter) |     76.5    |     76.2     |
>
> We find that the online version of SimPO continues to improve performance with each additional iteration.
>
> **Tasks other than chat, e.g., reasoning tasks like coding and maths**:
> In our initial manuscript (also shown in the table below), we did find that after further training on the Llama-3-Instruct-8B model, the chat ability of the models enhances at the sacrifice of the degradation on GSM8k (math) and MMLU (knowledge). However, after more exploration, we find that such a performance degradation is largely attributable to the lack of robustness of the base model (i.e., Llama-3) rather than the SimPO objective.
> We’d like to present the following new results from the gemma-2-9b-it model. The last two rows in the table below demonstrate that SimPO largely retains general knowledge (MMLU) and even slightly improves the original model’s math ability (GSM). Additionally, it significantly enhances coding ability, as demonstrated by the Arena-Hard benchmark, which primarily consists of real-world coding questions.
>
> |models | AlpacaEval 2 LC | AlpacaEval 2 WR | Arena-Hard  | GSM (0 shot) | MMLU (0 shot) |
> |-----|:------:|:------:|:----------:|:----:|:---------:|
> | Llama-3-Instruct-8B  |  26.0  |  25.3  |   22.3 |  78.5 |  61.7 |
> | Llama-3-Instruct-8B-SimPO |  44.7  |  40.5  |   33.8 |  71.3 |  58.5 |
> | gemma-2-9b-it  |  51.1  |  38.1  |   40.8 |  87.4 |  72.7 |
> | gemma-2-9b-it-SimPO |  72.4  |  65.9  |   59.1 |   88.0 |  72.2 |

---

> > ### Comment · Reviewer_TXhi · 2024-08-12
> > **Thanks**
> >
> > Thanks for the reply and I appreciate the authors' efforts.

---

> > > ### Author Response · Authors · 2024-08-12
> > > **Thank you for the support!**
> > >
> > > Dear Reviewer TXhi,
> > >
> > > Thank you once again for your thoughtful review and valuable feedback! We appreciate your support!
> > >
> > > Sincerely,
> > > Authors

---

### Official Review · Reviewer_oFKm · 2024-07-12

**Soundness:** 2
**Presentation:** 2
**Contribution:** 2
**Rating:** 3
**Confidence:** 4

**Summary:**

Simpo introduces a simplistic new method of alignment leveraging the log-probability of the sequence eliminating the need to leverage the SFT policy making the implementation computationally less complex. By introducing a notion of margin in the loss function, SimPO outperforms several existing alignment algorithms including DPO and variants without increasing response length. Overall, it provides an interesting light-weight alignment procedure with strong empirical benefits.

**Strengths:**

1. The approach eliminates the need for the reference model which can be computationally heavy or might not be available in practice
2. The methods shows improved empirical performance over DPO and existing baselines through extensive experimentations.
3. The method provides an interesting way of regularizing the length and shows improved performance without increasing the length of the response.

**Weaknesses:**

1. The primary definition of RLHF has a constraint to the SFT policy through the KL regularization which restricts the model to overoptimize the reward function. Also, it ensures the base performance of the LLM remains intact on several other domains and tasks. However, the proposed method doesn't use KL to the SFT and removes the SFT which makes it unclear what to attribute the improvement to. Specifically, one can violate the constraint and do better with the reward for that specific or similar task, but what about the performance in other tasks that the pre-trained/SFT model was good at?

2. DPO has a closed-form unique solution due to the strong convexity of the KL regularized RLHF objective resulting in the particular objective. However, it's not clear if we don't use the KL regularization, what the original optimization problem we are optimizing for. Is it still strongly convex? If not, what are the insights that this objective will have some good convergence properties?

3. Under all these questions, the motivation for the objective is heuristically driven and unclear from where it comes from

**Questions:**

1. Can you mention if your original objective is strongly convex or how the particular objective is derived?
2. Can you provide the KL regularization of your model to the SFT and show how much it deviates from the SFT in comparison to baselines?

**Limitations:**

Check above

---

> ### Author Rebuttal · Authors · 2024-08-07
>
> We'd like to thank the reviewer for the thoughtful feedback. We address your questions as follows.
>
> **Without KL to the SFT, one can violate the constraint and do better with the reward for that specific or similar task, but what about the performance in other tasks that the pre-trained/SFT model was good at?**
> While reward hacking is theoretically possible without explicit regularization, several practical factors mitigate this risk: (1) the learning rate, (2) the preference dataset, and (3) the SFT model’s robustness to forgetting.
> 1. The learning rate of preference optimization algorithms is typically very small (e.g., 5e-7), which naturally prevents significant divergence from the SFT model.
> 2. The preference datasets are often constructed to cover a broad range of tasks and domains, helping retain the model's existing knowledge and task versatility.
> 3. Large language models generally have a substantial capacity to learn from new data without catastrophic forgetting of previously learned tasks. This robustness can mitigate the need for explicit regularization, similar to how instruction tuning (SFT) retains the model’s pretrained knowledge without KL regularization.
>
> The above factors combined might be sufficient to ensure that the model learns human preferences while retains its generalization capabilities. Empirically, with appropriate hyperparameters, SimPO can result in a similar KL divergence to DPO (see our response to your last question) and comparable academic benchmark performance (see Table 9 in the Appendix). Additionally, we demonstrate new results in the table below that fine-tuning gemma2-9b-it with SimPO yields significantly improved instruction following ability without degradation on academic benchmarks including math (GSM) and general knowledge (MMLU).
>
> |models | AlpacaEval 2 LC | GSM (0 shot) | MMLU (0 shot) |
> |-----|:------:|:------:|:----------:|
> | gemma-2-9b-it  |  51.1  |  87.4 |  72.7 |
> | gemma-2-9b-it-DPO  |  67.8  |  88.5 |  72.2 |
> | gemma-2-9b-it-SimPO |  72.4  |  88.0 |  72.2 |
>
> **DPO has a closed-form unique solution due to the strong convexity of the KL regularized RLHF objective resulting in the particular objective. Without KL, is the objective still strongly convex? If not, what are the insights that this objective will have some good convergence properties?**
> The derivation of the DPO objective from RLHF relies on the specific condition of achieving the optimal policy $\pi^*_{\theta}$. However, this assumption is rarely met in practice due to the inherent challenges in optimizing deep neural networks. Therefore, there's no guarantee that DPO faithfully implements the original RLHF objective. Our work departs from this assumption: we directly address the potential misalignment between the DPO objective's reward metric and the decoding objective. This focus allows us to tackle the practical limitations of DPO without relying on idealized assumptions.
>
> Furthermore, comparing preference optimization (PO) objectives in terms of convexity or convergence is inherently difficult. All these objectives, including RLHF (PPO), DPO, and SimPO, are generally non-convex with respect to the model parameters $\theta$. This is because the policy model $\pi_{\theta}$ is parametrized by multi-layer, non-linear Transformer architectures, making it a non-convex function with respect to $\theta$. Consequently, any PO objective that depends on $\pi_{\theta}$ will also be inherently non-convex with respect to the model parameters.
>
> **The motivation for the objective is heuristically driven and unclear from where it comes from**:
> Our SimPO objective is systematically derived by aligning the reward in training with the generation likelihood in decoding:
> 1. We use the sequence likelihood objective, optimized by decoding algorithms, as the reward metric for winning/losing responses during preference optimization.
> 2. We then apply the Bradley-Terry objective over these reward metrics to formulate the SimPO objective.
> 3. The target reward margin hyperparameter in SimPO is analogous to the margin loss in SVMs and represents the home advantage in Bradley-Terry models.
>
> **Can you provide the KL regularization of your model to the SFT and show how much it deviates from the SFT in comparison to baselines?** In the PDF attached to our general response, we illustrate the KL divergence of SimPO and DPO. Figure 1 shows that (a) increasing $\beta$ in DPO can reduce the KL divergence, but (b) it can also overly constrain the model from effective learning of the preference dataset, resulting in lower generation quality. Therefore, there is a trade-off between learning preference data and staying close to the initial model. With appropriate hyperparameters, SimPO can achieve similar KL divergence to DPO.

---

> > ### Comment · Reviewer_oFKm · 2024-08-12
> > **Response to Rebuttal by Authors**
> >
> > Thanks for the points. I, in general, agree with the comment regarding the closed form of DPO and that it directly doesn't hold for a parametrized setting, which is ok. However, at least for the Tabular class of policies, it holds true and it indeed has the closed form. However, in your problem, how the optimization expression comes is not clear, once we remove the KL.
> > Also, can you explain why even after not having any KL constraint, SimPO can achieve similar KL divergence to DPO ? Which hyperparameters will cause it to stay close and why?

---

> > > ### Author Response · Authors · 2024-08-12
> > > **Response to Reviewer oFKm**
> > >
> > > Thank you for the reply! We address your questions as follows:
> > >
> > > **How the optimization expression comes is not clear, once we remove the KL**:
> > >
> > > While DPO is derived from the KL-regularized RLHF objective, it's important to highlight that this equivalence only holds at the "optimal policy" (i.e., $\pi_{\theta} = \pi^*$). However, this optimal condition is rarely, if ever, achieved during the actual training process due to the complexities of optimizing deep neural networks, thereby breaking the correspondence between RLHF and DPO in practice. Therefore, we believe that a full correspondence between RLHF and *PO is not the prerequisite for *PO algorithms to perform well empirically.
> > >
> > > The primary motivation behind SimPO is to **address the training-decoding discrepancy in DPO** which is a significant practical issue: In DPO’s objective, the implicit reward metric being optimized during training incorporates the reference model, which is absent during decoding. This mismatch can lead to a counterintuitive outcome: even if a winning response $y_w$ has a higher reward than a losing response $y_l$ during training (i.e., $r(x, y_w) > r(x, y_l)$), the model might still be more likely to generate $y_l$ during decoding, as illustrated in Figure 4 (b) in our manuscript.
> > >
> > > **Also, can you explain why even after not having any KL constraint, SimPO can achieve similar KL divergence to DPO? Which hyperparameters will cause it to stay close and why?**
> > >
> > > For *PO algorithms, the reference model is the initial SFT model where preference optimization begins (that’s why the KL divergence of DPO/SimPO always starts from 0). Two key hyperparameters in SimPO directly influence the KL divergence by controlling how much the model deviates from its initial state:
> > > * **Learning rate**: The learning rate of DPO/SimPO is typically very small (e.g., 5e-7), which naturally constrains model updates from the initial checkpoint, resulting in small KL divergence from the reference model. Increasing the learning rate can lead to greater KL divergence and performance degradation for both methods.
> > > * **The $\beta$ hyperparameter in the training objective**: The $\beta$ hyperparameter scales the reward difference in the SimPO objective: $\mathcal{L} = -\log \sigma  \left( \frac{\beta}{|y_w|} \log \pi_\theta(y_w|x) - \frac{\beta}{|y_l|} \log \pi_\theta(y_l|x) - \gamma \right)$. A large $\beta$ amplifies even small differences between the winning likelihood $\frac{\beta}{|y_w|} \log \pi_\theta(y_w|x)$ and the losing likelihood $\frac{\beta}{|y_l|} \log \pi_\theta(y_l|x)$. Consequently, even a slight advantage of the winning response will lead to a near-zero loss, minimizing model updates and keeping the KL divergence small.
> > >
> > > In practice, we observe that for both DPO and SimPO, hyperparameters can significantly impact the results, further supporting the point that deriving from RLHF does not guarantee empirical robustness.

---

> > > ### Author Response · Authors · 2024-08-13
> > >
> > > Dear reviewer oFKm,
> > >
> > > As the discussion period is coming to an end, we kindly ask that you review our responses and let us know if you have any additional concerns. If possible, we would appreciate it if you could adjust your scores accordingly. Thank you!

---

### Official Review · Reviewer_a8Vq · 2024-07-14

**Soundness:** 3
**Presentation:** 3
**Contribution:** 3
**Rating:** 7
**Confidence:** 4

**Summary:**

The paper introduces SimPO (Simple Preference Optimization), an extension of Direct Preference Optimization (DPO), by replacing the reference-policy-dependent implicit reward with a reference-free reward.
Specifically, SimPO utilizes the average log probability of a sequence as the implicit reward, aligning it with the generation process and eliminating the need for a reference model. This results in improved computational and memory efficiency. Additionally, a target reward margin is introduced into the Bradley-Terry objective to ensure a significant reward difference between winning and losing responses. Extensive experiments show that SimPO consistently outperforms DPO and its variants across various benchmarks, demonstrating its effectiveness in improving model performance.

**Strengths:**

- The focus on addressing the discrepancy between reward and generation in DPO is novel and valuable, providing a new perspective on extending DPO.
- SimPO enhances computational and memory efficiency by eliminating the need for a reference model.
- Extensive experiments across various benchmarks demonstrate that SimPO consistently outperforms DPO and its variants.

**Weaknesses:**

- Although SimPO simplifies the optimization process by removing the reference model calculation of response probabilities, it introduces a new hyperparameter, the target reward margin $\gamma$, which requires tuning.
- While the motivation to address the discrepancy between reward and generation in DPO is understandable, there is a lack of mathematical discussion on how the proposed implicit reward is effective for preference optimization. For instance, standard random sampling with temperature 1 could eliminate the discrepancy without length normalization. However, the experimental results suggest that length normalization is crucial, questioning whether the initial motivation to align optimization and decoding is definitively effective for preference optimization.
- The target reward margin plays a critical role in SimPO, but there is no discussion or comparative experiments with "DPO with an offset" (Amini et al., 2024), which also introduces a similar margin into the DPO loss. This comparison is necessary to understand the relative advantages of SimPO.

**Questions:**

- Is it possible to add a careful discussion and conduct comparative experiments about DPO with a target reward margin (Amini et al., 2024)?
- The paper suggests that reducing the discrepancy between reward and generation in DPO results in a more preferable reward for preference optimization. While theoretical analysis might be challenging, is it possible to experimentally validate this hypothesis? For instance, can you evaluate the relationship between the discrepancy's size and preference optimization's performance? This could include testing different decoding methods, such as a beam search with larger beam widths in the length-normalized version that will align more with the average log-likelihood reward.
- Since SimPO omits KL divergence regularization from the initial policy, assessing how the KL divergence behaves would be helpful. Evaluating the model's performance in terms of KL divergence, as analyzed by Gao et al. (2023), could provide valuable insights.

**Limitations:**

The authors have adequately discussed the limitations.

---

> ### Author Rebuttal · Authors · 2024-08-07
>
> We’d like to thank the reviewer for acknowledging the novelty and simplicity of our proposed approach. We address your raised points as follows.
>
> **The target reward margin $\gamma$ requires extra tuning**:
> We acknowledge that the newly introduced target reward margin requires additional tuning. However, we’d like to emphasize that within a reasonable range of $\gamma$ values, SimPO consistently outperforms the DPO baseline, as shown in the table below (also Figure 3a in the manuscript). Furthermore, we find that using $\gamma$=1.5 generally yields sufficiently good results across all settings. While slight tuning can further improve performance, it is not mandatory.
> | Models           | AlpacaEval LC Win Rate  |
> |------------------|:-------------------------:|
> | SimPO ($\gamma$=0)   |  16.8 |
> | SimPO ($\gamma$=0.8) |  20.2 |
> | SimPO ($\gamma$=1.6) |  22.0 |
> | SimPO ($\gamma$=2.4) |  16.8 |
> | DPO  | 15.1 |
>
> **Lack of mathematical understanding**:
> While it will be challenging to derive a rigorous theoretical understanding yet, we offer two possible explanations:
> 1. SimPO's reward, the average log-likelihood of a sequence, closely aligns with the objective used during decoding. Although it's challenging to prove rigorously, random sampling with a temperature of 1 is likely to match SimPO's reward metric better than either (1) DPO or (2) SimPO without length normalization.
>     * DPO involves a reference model in its formulation, which is not used during decoding.
>     * Analogous to beam search, which ranks candidate sequences of varying lengths using average log-likelihood, length normalization is crucial when comparing sequences of different lengths (i.e., winning and losing responses) for reward calculation in SimPO. The reward without length normalization is biased, as evidenced by the tendency to assign higher likelihoods to longer sequences (Figure 2). This can result in excessively long outputs, such as degeneration into repetitive tokens.
> 2. Another perspective for understanding the method is that the SimPO reward decouples the reward from the sequence length. According to [1], sequence length positively correlates with model-based judgment. SimPO’s disentanglement allows us to train the policy without biasing it to generate longer sequences merely to achieve a higher reward. Instead, it encourages the model to learn and focus on the differences beyond sequence lengths.
>
> We hope this helps clarify the effectiveness of our approach!
>
> [1] [A Long Way to Go: Investigating Length Correlations in RLHF](https://arxiv.org/abs/2310.03716)
>
> **DPO with an offset (Amini et al., 2024)**: We conducted additional experiments to determine if adding an offset (Amini et al., 2024) would further enhance DPO. After tuning $\gamma$, we found that it does not lead to any further improvement. Specifically, we used $\gamma$=0.1 for mistral-base and $\gamma$=0.0 for mistral-instruct. Increasing gamma beyond these values resulted in worse results.
> |              |    |         Mistral-base 7B     |            |   |       Mistral-instruct 7B       |            |
> |--------------|:---------------:|:------------:|:----------:|:-------------------:|:------------:|:----------:|
> |              | AlpacaEval 2 LC Win Rate | AlpacaEval 2 Win Rate | Arena-Hard |   AlpacaEval 2 LC Win Rate   | AlpacaEval 2 Win Rate | Arena-Hard |
> | DPO          |       15.1      |     12.5     |    10.4    |         26.8        |     24.9     |    16.3    |
> | DPO w/ $\gamma$ |       15.2      |     12.1     |    10.3    |       26.8        |     24.9     |    16.3   |
>
> The objective of DPO inherently includes an instance-wise target reward margin $\gamma_{\text{ref}}$, as shown below:
> $$
> \mathcal{L} = \log \sigma \left( \beta \log \frac{\pi_\theta(y_w \mid x)}{\pi_{\text{ref}}(y_w \mid x)} - \beta \log \frac{\pi_\theta(y_l \mid x)}{\pi_{\text{ref}}(y_l \mid x)}\right)\\
> = \log \sigma \bigg( \beta \log \pi_\theta(y_w \mid x) - \beta \log \pi_\theta(y_l \mid x) - \big(\beta \log \pi_{\text{ref}}(y_w \mid x) - \beta \log \pi_{\text{ref}}(y_l \mid x)\big)\bigg)
> $$
> where
> $\gamma_{\text{ref}} = \beta \log \pi_{\text{ref}}(y_w \mid x) - \beta \log \pi_{\text{ref}}(y_l \mid x)$.
> This may explain why adding an extra margin to DPO will not be as effective as it is with SimPO. We will add these results and discussions to our revision!
>
> **Discuss about KL divergence**:
> In the PDF attached to our general response, we illustrate the KL divergence of SimPO and DPO across different $\beta$, measured on the winning responses from a held-out set during training. In summary:
> * We find that SimPO, even without KL regularization, yields a reasonably small KL divergence from the policy model to the SFT model, which is comparable to DPO under its optimal $\beta$.
> * Within the range of $\beta$ explored, increasing $\beta$ reduces the KL divergence for both DPO and SimPO, but it can also overly constrain the model from effective learning of the preference dataset, resulting in lower generation quality. This suggests that KL divergence is not monotonically related to reward scores or model quality (Gao et al., 2023 present a similar phenomenon).
> We hope this helps clarify the reviewer’s concern about KL divergence!

---

> ### Author Response · Authors · 2024-08-07
> **Explore beam search with a large beam**
>
> Thanks for the insightful suggestion to test the discrepancy between the reward metric and decoding methods! We follow the reviewer's suggestion to test beam search with different beam sizes (e.g., 1, 5, 10) using the mistral-base-SimPO model. We find that increasing beam sizes does increase the generation quality measured by win rate, as shown in the following table. It's even better than the number reported in our paper! We think that this helps validating that aligning rewards and decoding metrics more explicitly is beneficial. However, a clear downside is that the runtime significantly increases as beam size increases, and we think this might be why people stick with greedy decoding/sampling instead of using beam search for LLM generation.
>
> | Decoding Method | LC Win Rate |  WR  |   runtime   |
> |:---------:|:-----------:|:----:|:--------:|
> |  sampling |     21.5    | 20.8 |   5min   |
> |     greedy (beam=1)     |     22.6    | 22.3 |   5min   |
> |     beam=5     |     22.2    | 21.6 | 1h 51min |
> |     beam=10    |     24.5    | 23.5 | 3h 56min |

---

> > ### Author Response · Authors · 2024-08-12
> > **Looking forward to your feedback!**
> >
> > Dear Reviewer a8Vq,
> >
> > Thank you once again for your thoughtful review and valuable feedback! As the discussion period is ending tomorrow, we would greatly appreciate knowing whether our response has adequately addressed your questions.
> >
> > If you have any additional comments or concerns, please feel free to share them with us.
> >
> > Sincerely,
> > Authors

---

> > > ### Comment · Reviewer_a8Vq · 2024-08-13
> > >
> > > Thanks to the author(s) for clarifying the concerns. The additional experiments are helpful in analyzing SimPO more deeply. However, I am confused about the mathematical understanding regarding the role of length normalization. In the context of beam search, length normalization is typically used to address the bias toward generating shorter sequences, which arises because the log probabilities (which are always negative since the max probability is 1) tend to favor shorter sequences. This adjustment prevents the model from preferring short sequences excessively (e.g., [A]).
> > >
> > > However, the statement in the paper (“removing the length normalization term from the reward formulation results in a bias toward generating longer but lower-quality sequences,” in line 101) and response (“The reward without length normalization is biased, as evidenced by the tendency to assign higher likelihoods to longer sequences”) seem to contradict the conventional understanding of length normalization. Could the authors please clarify this apparent inconsistency?
> > >
> > > On the second point ("Another perspective for ... "), I appreciate the explanation that SimPO’s reward decouples the sequence length from the reward, allowing the model to focus on meaningful differences beyond just the length.
> > >
> > > [A] Kenton Murray, David Chiang: “Correcting Length Bias in Neural Machine Translation,” ACL 2018. (https://aclanthology.org/W18-6322/)

---

> ### Author Response · Authors · 2024-08-13
> **Reponse to Reviewer a8Vq**
>
> Thank you for the thoughtful comments; this is an excellent question! To clarify how length normalization (LN) impacts preference learning, we need to consider three factors:
> - **Length bias of total log-likelihood**: As you mentioned, longer sequences typically accumulate a lower (more negative) summed log-likelihood, inherently disadvantaging longer sequences.
> - **Length difference in preference pairs**: We found that, in many instances, the winning response ($y_w$) is longer than the losing response ($y_l$).
> - **Objective of preference learning**: Preference learning aims to assign higher rewards to winning responses ($y_w$) compared to losing ones ($y_l$).
>
> Without LN, the reward metric $r_{\text{SimPO w/o LN}}(x,y) = \log \pi_\theta(y \mid x)$ is based on the total log-likelihood. Due to the length bias, longer winning responses ($y_w$) generally have a lower total log-likelihood compared to shorter losing responses ($y_l$). To correctly rank these $y_w$ higher, the model must overcome this bias by assigning disproportionately high probabilities to each token in $y_w$ to ensure the reward on $y_w$ exceeds that on $y_l$. This compensatory effect can cause the model to become miscalibrated, as illustrated in Figure 2(c), where it begins to excessively favor longer sequences. As a result, during decoding, such an ill-calibrated model tends to generate longer sequences. Essentially, LN acts as a calibration mechanism to prevent this overcompensation in the training stage.
>
> To further illustrate this for a mathematical understanding, let's analyze the loss of SimPO with and without LN. For simplicity, we omit the expectation symbol in the following formulas:
>
> - With LN: $ L_{\text{SimPO}} (\pi_\theta) = - \log \sigma  \left( \frac{\beta}{|y_w|} \log \pi_\theta(y_w|x) - \frac{\beta}{|y_l|} \log \pi_\theta(y_l|x) - \gamma \right)$
> - Without LN: $ L_{\text{SimPO w/o LN}} (\pi_\theta) = - \log \sigma  \left( \log \pi_\theta(y_w|x) - \log \pi_\theta(y_l|x) - \gamma \right)$
>
> It can be seen that without LN, the reward difference $\Delta r = \log \pi_\theta(y_w|x) - \log \pi_\theta(y_l|x)$ is based on **total log-likelihoods**, and is generally more negative when $y_w$ is longer than $y_l$ due to the length bias. Consequently, $\sigma(\Delta r - \gamma)$ approaches 0, and thus the final loss for such instances, $- \log \sigma(\Delta r - \gamma)$, will be very large. This heavily biases the model towards learning such preference pairs. As demonstrated in Figure 2(a), this can result in the failure of learning the opposite cases where $y_w$ is shorter than $y_l$ (the loss for these cases will be very small by applying a similar reasoning).
>
> We hope this addresses the reviewer's question, and we’ll include these clarifications in the revision. If you find this explanation satisfactory, we kindly ask you to consider raising the score, thank you so much for your time! We are also happy to help further clarify any questions you have!

---

> > ### Comment · Reviewer_a8Vq · 2024-08-14
> >
> > Thanks for the reply. The explanations provided were convincing and significantly contributed to a deeper understanding of SimPO. I appreciate the detailed discussion and would encourage careful inclusion of these points in the revised paper. I have also reviewed the other reviewers' comments and the corresponding responses. I have no additional concerns, so I have raised the score by one level.

---

### Author Rebuttal · Authors · 2024-08-07

We sincerely thank all reviewers for their thoughtful feedback, and we'd like to share some additional analysis and results since the submission!

**Additional Analysis in the attached PDF**
We include the following additional studies:
* KL divergence plots of SimPO vs. DPO
* Qualitative studies of SimPO vs. DPO

**Exceptionally strong results from applying SimPO to gemma-2-9b-it**
Furthermore, we’d like to present our new results from training gemma-2-9b-it with SimPO. The resulting model tops the AlpacaEval 2 leaderboard and the Arena-Hard benchmark among similar-sized models. Importantly, we find that training the gemma model with SimPO retains its knowledge (MMLU) and even slightly improves the original model’s math ability (GSM). This exciting new set of results demonstrates the effectiveness of SimPO across different model types without performance degradation on other benchmarks.

|models | AlpacaEval 2 LC | AlpacaEval 2 WR | Arena-Hard  | GSM (0 shot) | MMLU (0 shot) |
|-----|:------:|:------:|:----------:|:----:|:---------:|
| gemma-2-9b-it  |  51.1  |  38.1  |   40.8 |  87.4 |  72.7 |
| gemma-2-9b-it-SimPO |  72.4  |  65.9  |   59.1 |   88.0 |  72.2 |

---

### Decision · Program_Chairs · 2024-09-25

**Decision:**

Accept (poster)

**Comment:**

This paper proposes SimPO for preference optimization without using a reference model. The two key designs are length-normalized reward and target reward margin. Experiments on AlpacaEval, MT-Bench and Arena-Hard verify the proposed method. Reviewers raised concerns about why and how without KL regularization SimPO is able to preserve close enough distance between trained and reference policies, as well as more experimental results, and the authors addressed most of the concerns. I suggest that the authors incorporate the discussions and additional experiments in subsequent versions of the paper.